

**Realized ecological forecast through interactive Ecological Platform for Assimilating Data**
**into model (EcoPAD)**
Yuanyuan Huang[1,2], Mark Stacy[3], Jiang Jiang[1,4], Nilutpal Sundi[5], Shuang Ma[1,6], Volodymyr
Saruta[1,6], Chang Gyo Jung[1,6], Zheng Shi[1], Jianyang Xia[7,8], Paul J.Hanson[9], Daniel Ricciuto[9], Yiqi
Luo[1,6,10]
1, Department of Microbiology and Plant Biology, University of Oklahoma, Norman, Oklahoma
2, Laboratoire des Sciences du Climat et de l'Environnement, 91191 Gif-sur-Yvette, France
3, University of Oklahoma Information Technology, Norman, Oklahoma, USA
4, Key Laboratory of Soil and Water Conservation and Ecological Restoration in Jiangsu Province,
Collaborative Innovation Center of Sustainable Forestry in Southern China of Jiangsu Province, Nanjing
Forestry University, Nanjing, Jiangsu, China
5, Department of Computer Science, University of Oklahoma, Norman, Oklahoma, USA
6, Center for Ecosystem Science and Society, Northern Arizona University, Flagstaff, AZ, USA
7, Tiantong National Forest Ecosystem Observation and Research Station, School of Ecological and
Environmental Sciences, East China Normal University, Shanghai 200062, China.
8, Research Center for Global Change and Ecological Forecasting, East China Normal
University, Shanghai 200062, China
9, Environmental Sciences Division and Climate Change Science Institute, Oak Ridge National
Laboratory, Oak Ridge, Tennessee, USA
10, Department of Earth System Science, Tsinghua University, Beijing 100084 China
Correspondence: Yuanyuan Huang (yuanyuanhuang2011@gmail.com) and Yiqi Luo
(Yiqi.Luo@nau.edu)



**Abstract.** Predicting future changes in ecosystem services is not only highly desirable but also
becomes feasible as several forces (e.g., available big data, developed data assimilation (DA)
techniques, and advanced cyberinfrastructure) are converging to transform ecological research to
quantitative forecasting. To realize ecological forecasting, we have developed an Ecological
Platform for Assimilating Data (EcoPAD) into models. EcoPAD is a web-based software system
that automates data transfer and processes from sensor networks to ecological forecasting
through data management, model simulation, data assimilation, and visualization. It facilitates
interactive data-model integration from which model is recursively improved through updated
data while data is systematically refined under the guidance of model. EcoPAD relies on data
from observations, process-oriented models, DA techniques, and web-based workflow.

We applied EcoPAD to the Spruce and Peatland Responses Under Climatic and

Environmental change (SPRUCE) experiment at North Minnesota. The EcoPAD-SPRUCE
realizes fully automated data transfer, feeds meteorological data to drive model simulations,
assimilates both manually measured and automated sensor data into Terrestrial ECOsystem
(TECO) model, and recursively forecast responses of various biophysical and biogeochemical
processes to five temperature and two $CO_2$ treatments in near real-time (weekly). The near real-
time forecasting with EcoPAD-SPRUCE has revealed that uncertainties or mismatches in
forecasting carbon pool dynamics are more related to model (e.g., model structure, parameter,
and initial value) than forcing variables, opposite to forecasting flux variables. EcoPAD-
SPRUCE quantified acclimations of methane production in response to warming treatments
through shifted posterior distributions of the $CH_4$:$CO_2$ ratio and temperature sensitivity ($Q_{10}$) of
methane production towards lower values. Different case studies indicated that realistic
forecasting of carbon dynamics relies on appropriate model structure, correct parameterization



and accurate external forcing. Moreover, EcoPAD-SPRUCE stimulated active feedbacks
between experimenters and modelers so as to identify model components to be improved and
additional measurements to be made. It becomes the first interactive model-experiment (ModEx)
system and opens a novel avenue for interactive dialogue between modelers and experimenters.

EcoPAD also has the potential to become an interactive tool for resource management, to

stimulate citizen science in ecology, and transform environmental education with its easily
accessible web interface.
**Key words:**
Data assimilation, SPRUCE, carbon, global change, real time, acclimation, forecast



## 1. Introduction

One ambitious goal of ecology as one science discipline is to forecast future states and
services of ecological systems. Forecasting futures in ecology is not only desirable for scientific
advances in this discipline but also has practical values to guide resource management and
decision-making toward a sustainable planet earth. The practical need for ecological forecasting
is particularly urgent in this rapidly changing world, which is experiencing unprecedented
natural resource depletion, increasing food demand, serious biodiversity crisis, accelerated
climate changes, and widespread pollutions in the air, waters, and soils [*Clark et al.*, 2001;
*Mouquet et al.*, 2015]. As a result, a growing number of studies have been reported in the last
several decades on forecasting of phenology [*Diez et al.*, 2012], carbon dynamics [*Gao et al.*,
2011; *Luo et al.*, 2016; *Thomas et al.*, 2017], species dynamics [*Clark et al.*, 2003; *Kearney et*
*al.*, 2010], pollinator performance[*Corbet et al.*, 1995], epidemics [*Ong et al.*, 2010], fishery
[*Hare et al.*, 2010], algal bloom [*Stumpf et al.*, 2009], crop yield [*Bastiaanssen and Ali*, 2003],
biodiversity [*Botkin et al.*, 2007], plant extinction risk [*Fordham et al.*, 2012], and ecosystem
service [*Craft et al.*, 2009]. Despite its broad applications, ecological forecasting is still
sporadically practiced and lags far behind demand due to the lack of infrastructure that enables
timely integration of models with data. This paper introduces the fully interactive infrastructure,
the E̲co̲logical P̲latform for A̲ssimilating D̲ata (EcoPAD) into models, to inform near-time
ecological forecasting with iterative data-model integration.
Ecological forecasting relies on both models and data. However, currently the ecology
research community has not yet adequately integrated observations with models to inform best
forecast. Forecasts generated from scenario approaches are qualitative and scenarios are often
not based on ecological knowledge [*Coreau et al.*, 2009; *Coreau et al.*, 2010]. Data-driven



forecasts using statistical methods are generally limited for extrapolation and sometimes
contaminated by confounding factors [*Schindler and Hilborn*, 2015]. Recent emergent
mechanism-free non-parametric approach, which depends on the statistical pattern extracted
from data, is reported to be promising for short-term forecast [*Sugihara et al.*, 2012; *Perretti et
al.*, 2013; *Ward et al.*, 2014], but has limited capability in long-term prediction due to the lack of
relevant ecological mechanisms. Process-based models provide the capacity in long term
prediction and the flexibility in capturing short term dynamics on the basis of mechanistic
understanding [*Coreau et al.*, 2009; *Purves et al.*, 2013]. Wide applications and tests of process-
based models are limited by their often complicated numerical structure and sometimes
unrealistic parameterization [*Moorcroft*, 2006]. The complex and uncertain nature of ecology
precludes practice of incorporating as many processes as possible into mechanistic models. Our
current incomplete knowledge about ecological systems or unrepresented processes under novel
conditions is partly reflected in model parameters which are associated with large uncertainty.
Good forecasting therefore requires effective communication between process-based models and
data to estimate realistic model parameters and capture context-dependent ecological
phenomena.

Data-model fusion, or data-model integration, is an important step to communicate model

with data. But previous data-model integration actvities have mostly been done in an *ad hoc*
manner instead of being interactive. For example, data from a network of eddy covariance flux
tower sites across United States and Canada was compared with gross primary productivity
(GPP) estimates from different models [*Schaefer et al.*, 2012]. *Luo and Reynolds* [1999] used a
model to examine ecosystem responses to gradual as in the real world vs. step increases in $CO_2$
concentration as in elevated $CO_2$ experiments. *Parton et al.* [2007] parameterized $CO_2$ impacts in





an ecosystem model with data from a $CO_2$ experiment in Colorado. Such model-experiment
interactions encounter a few issues: 1) Models are not always calibrated for individual sites and,
therefore, not accurate; 2) It is not very effective because it is usually one-time practice without
many iterative processes between experimenters and modelers [*Dietze et al.*, 2013; *Lebauer et*
*al.*, 2013]; 3) It is usually one-directionary as data is normally used to train models while the
guidance of model for efficient data collection is limited; and 4) It is not streamlined and could
not be disseminated with common practices among the research community [*Dietze et al.*, 2013;
*Lebauer et al.*, 2013; *Walker et al.*, 2014].
A few research groups have developed data assimilation systems to faciliate data-model
integration in a systematic way. For example, data-model integration systems, such as the Data
Assimilation Research Testbed - DART [*Anderson et al.*, 2009], the General Ensemble
Biogeochemical Modeling System - GEMS [*Tan et al.*, 2005] and the Carbon Cycle Data
Assimilation Systems - CCDAS [*Scholze et al.*, 2007; *Peylin et al.*, 2016], combine various data
streams (e.g., FLUXNET data, satellite data and inventory data) with process-based models
through data assimilation algorithms such as the Kalman filter [*Anderson et al.*, 2009] and
variational methods [*Peylin et al.*, 2016]. These data assimilaiton systems automate model
parameterization and provided an avenue to systematically improve models through combining
as much data as possible. Model improvements normally happen after the ending of an field
experiment and the interactive data-model intergration is limited as feedbacks from models to
ongoing experimetal studies are not adequately realized. In addtion, wide applications of these
data assimilation systems in ecological forecasting are constrained by limited user interactions
with its steep learning curve to understand these systems, especially for exmperimenters who
have limited training in modeling.



Realizing interactive ecological forecasting requires web-based technology to faciliate
scientific workflow, the sequence of processes through which a piece of work passes from
initiation to completion. Web-based modeling, which provides user-friendly interfaces to run
models in the background, is uaully supported by scientific workflow. For example,
TreeWatch.Net has recently been developed to make use of  high precision individual tree
monitoring data to parameterize process based tree models in real-time and to assess instant tree
hydraulics and carbon status with online result visualization [*Steppe et al.*, 2016]. Although the
web portal of TreeWatch.Net is currently limited to visualization purposes, it largely broadens
the application of data-model integration and strengthens the interaction of modeling results with
the general public. The Predictive Ecosystem Analyzer (PEcAn) is a scientific workflow that
wraps around different ecosystem models and manages the flows of information coming in and
out of the model [*Lebauer et al.*, 2013]. PEcAn enables web-based model similations. Such a
workflow has advantages, for exmaple, making ecological modeling and analysis convenient,
transparent, reproducible and adaptable to new questions [*Lebauer et al.*, 2013], and encouraging
user-model interactions. PEcAn uses the Bayesian meta-analysis to synthesize plant trait data to
estimate model parameters and associated uncertanties. Parameter uncertainties are propogated
to model uncertanties and displayed as outputs. It is still not fully interactive in the way that
states are not updated iteractively according to observations and the web-based data assimilation
and then ecoloical forecasting have not yet been fully realized.
The iterative model-data intergration is an important step to realize real or near real-time
ecological forecasting. Instead of projecting into future only one time through assimulating
available observations, interactive forecasting constantly updates forecasting as soon as new data
stream arrives or/and model is modified. Forecasting is likely to be improved unidirectionally in



which models are constantly updated through observations, or data collections/field
experimentations are regularly improved according to theoretical/model information. Ecological
forecasting can also be bidirectionally improved so that both models and field experimetations
are optimized hand in hand over time. Although the bidirctional case is rare in ecological
forecasting, the unidirectional iterative forecasting has been reported. One excellent example of
forecasting through dynamically and repeatedly integrating data with models is from infectious
disease studies [*Ong et al.*, 2010; *Niu et al.*, 2014]. Dynamics of infectious diseases are
tranditionaly captured by Susceptible-Infected-Removed (SIR) models. In the forecasting of the
Singapore H1N1-2009 infections, SIR model parameters and the number of individuals in each
state were updated daily, combining data renewed from local clinical reports. The evolving of the
epidemic related parameters and states were captured through iteratively assimilating
observations to inform forecasting. As a result, the model correctly forecasted the timing of the
peak and declining of the infection ahead of time. Iterative forecasting dynamically integrates
data with model and makes best use of both data and theoretical understandings of ecological
processes.

The aim of this paper is to present a fully interactive platform, a web-based Ecological

Platform for Assimilating Data into models (EcoPAD), to best inform ecological forecasting.
The interactive feature of EcoPAD is reflected in the iterative model updating and forecasting
through dynamically integrating models with new observations, bidirectional feedbacks between
experimenters and modelers, and flexible user-model communication through web-based
simulation, data assimilation and forecasting. Such an interactive platform provides the
infrastructure to effectively integrate available resources, from both models and data, modelers
and experimenters, scientists and the general public, to improve scientific understanding of



ecological processes, to boost ecological forecasting practice and transform ecology towards
qualitative forecasting.

In the following sections, we first describe the system design, major components and

functionality of EcoPAD. We then use the Spruce and Peatland Responses Under Climatic and
Environmental change (SPRUCE) experiment [*Hanson et al.*, 2017] as a testbed to elaborate new
opportunities brought by the platform. We finally discuss implications of EcoPAD for better
ecological forecasting.

**2 EcoPAD: system design, components, and functionality**
**2.1 General description: web-based data assimilation and forecast**

EcoPAD (https://ecolab.nau.edu/ecopad_portal/) focuses on linking ecological

experiments/data with models and allows easily accessible and reproducible data-model
integration with interactive web-based simulation, data assimilation and forecast capabilities.
Specially, EcoPAD enables the automated near time ecological forecasting which works hand-in-
hand between modelers and experimenters and updates periodically in a manner similar to the
weather forecasting. The system is designed to streamline web request-response, data
management, modeling, prediction and visualization to boost the overall throughput of
observational data, promote data-model communication, inform ecological forecasting and
improve scientific understanding of ecological processes.

To realize such data-informed ecological forecasting, the essential components of

EcoPAD include experiments/data, process-based models, data assimilation techniques and the
scientific workflow (Figures 1-3). The scientific workflow of EcoPAD that wraps around
ecological models and data assimilation algorithms acts to move datasets in and out of structured





and cataloged data collections (metadata catalog) while leaving the logic of the ecological
models and data assimilation algorithms untouched (Figures 1, 3). Once a user makes a request
through the web browser or command line utilities, the scientific workflow takes charge of
triggering and executing corresponding tasks, be it pulling data from a remote server, running a
particular ecological model, automating forecasting or making the result easily understandable to
users (Figures 1, 3). With the workflow, the system is agnostic to operation system, environment
and programming language and is built to horizontally scale to meet the demands of the model
and the end user community.

**2.2 Components**
**2.2.1 Data**
Data is an important component of EcoPAD and EcoPAD offers systematic data management to
digest diverse data streams. The 'big data' ecology generates plethora of very different datasets
across various scales [*Hampton et al.*, 2013; *Mouquet et al.*, 2015]. These datasets might have
high temporal resolutions, such as those from real time ecological sensors, or the display of
spatial information from remote sensing sources and data stored in the geographic information
system (GIS). These datasets may also include, but are not limited to, inventory data, laboratory
measurements, FLUXNET databases or from long term ecological networks. Such data contain
information related to environmental forcing (e.g., precipitation, temperature and radiative
forcing), site characteristics (including soil texture, species composition) and biogeochemical
information. Datasets in EcoPAD are derived from other research projects in comma separated
value files or other loosely structured data formats. These datasets are first described and stored
with appropriate metadata via either manual operation or scheduled automation from sensors.



Attention is then spent on how the particular dataset varies over space (x, y) and time (t). When
the spatiotemporal variability is understood, it is then placed in metadata records that allow for
query through its scientific workflow.
**2.2.2 Ecological models**

Process-based ecological model is another essential component of EcoPAD (Figure 1). In

this paper, the Terrestrial ECOsystem (TECO) model is applied as a general ecological model for
demonstration purpose since the workflow and data assimilation system of EcoPAD are
independent on the specific ecological model. TECO simulates ecosystem carbon, nitrogen,
water and energy dynamics [*Weng and Luo*, 2008; *Shi et al.*, 2016]. The original TECO model
has 4 major submodules (canopy, soil water, vegetation dynamics and soil carbon/nitrogen) and
is further extended to incorporate methane biogeochemistry and snow dynamics [*Huang et al.*,
2017; *Ma et al.*, 2017]. As in the global land surface model CABLE [*Wang and Leuning*, 1998;
*Wang et al.*, 2010], canopy photosynthesis that couples surface energy, water and carbon fluxes
is based on  a two-big-leaf model [*Wang and Leuning*, 1998]. Leaf photosynthesis and stomatal
conductance are based on the common scheme from *Farquhar et al.* [1980] and *Ball et al.* [1987]
respectively. Transpiration and associated latent heat losses are controlled by stomatal
conductance, soil water content and the rooting profile. Evaporation losses of water are balanced
between the soil water supply and the atmospheric demand which is based on the difference
between saturation vapor pressure at the temperature of the soil and the actual atmospheric vapor
pressure. Soil moisture in different soil layers is regulated by water influxes (e.g., precipitation
and percolation) and effluxes (e.g., transpiration and runoff). Vegetation dynamic tracks
processes such as growth, allocation and phenology. Soil carbon/nitrogen module tracks carbon
and nitrogen through processes such as litterfall, soil organic matter (SOM) decomposition and



mineralization. SOM decomposition modeling follows the general form of the Century model
[*Parton et al.*, 1988] as in most earth system models in which SOM is divided into pools with
different turnover times (the inverse of decomposition rates) which are modified by
environmental factors such as the soil temperature and moisture.
**2.2.3 Data assimilation**
Data assimilation is a cutting-edge statistical approach that integrates data with model in
a systematical way (Figure 2). Data assimilation is growing in importance as the process based
ecological models, despite largely simplifying the real systems, are in great need to be complex
enough to address sophisticate ecological issues that are composed of an enormous number of
biotic and abiotic factors interacting with each other. Data assimilation techniques provide a
framework to combine models with data to estimate model parameters [*Shi et al.*, 2016], test
alternative ecological hypotheses through different model structures [*Liang et al.*, 2015], assess
information content of datasets [*Weng and Luo*, 2011], quantify uncertainties [*Weng et al.*, 2011;
*Keenan et al.*, 2012; *Zhou et al.*, 2012], identify model errors and improve ecological predictions
[*Luo et al.*, 2011b]. Under the Bayesian paradigm, data assimilation techniques treat the model
structure, initial and parameter values as priors that represent our current understanding of the
system. As new information from observations or data becomes available, model parameters and
state variables can be updated accordingly. The posterior distributions of estimated parameters or
state variables are imprinted with information from both the model and the observation/data as
the chosen parameters act to reduce mismatches between observations and model simulations.
Future predictions benefit from such constrained posterior distributions through forward
modeling (Figure A1). As a result, the probability density function of predicted future states



through data assimilation normally has a narrower spread than that without data assimilation
when everything else is equal [*Luo et al.*, 2011b; *Weng and Luo*, 2011; *Niu et al.*, 2014].

EcoPAD is open to different data assimilation techniques depending on the ecological

questions under study since the scientific workflow of EcoPAD is independent on the specific
data assimilation algorithm. For demonstration, the Markov chain Monte Carlo (MCMC) [*Xu et*
*al.*, 2006] is described in this study.

MCMC is a class of sampling algorithms to draw samples from a probability distribution

obtained through constructed Markov Chain to approximate the equilibrium distribution, which
makes Bayesian inference, especially these with multi-dimensional integrals, workable. The
Bayesian based MCMC method is advantageous for better ecological forecasting as it takes into
account various uncertainty sources which are crucial in interpreting and delivering forecasting
results [*Clark et al.*, 2001]. In the application of MCMC, the posterior distribution of parameters
for given observations is proportional to the prior distribution of parameters and the likelihood
function which is linked to the fit/match (or cost function) between model simulations and
observations. EcoPAD currently adopts a batch mode, that is, the cost function is treated as a
single function to be minimized and different observations are standardized by their
corresponding standard deviations [*Xu et al.*, 2006]. For simplicity, we assume uniform
distributions in priors, and Gaussian or multivariate Gaussian distributions in observational
errors, which can be easily expanded to other specific distribution forms depending on the
available information. Detailed description is available in *Xu et al.* [2006].
**2.2.4 Scientific workflow**

EcoPAD relies on its scientific workflow to interface ecological models and data

assimilation algorithms, managing diverse data streams, automates iterative ecological



forecasting in response to various user requests. Workflow is a relatively new concept in the
ecology literature but essential to realize real or near-real time forecasting. Thus, we describe it
in details below. The essential components of a scientific workflow of EcoPAD include the
metadata catalog, web application-programming interface (API), the asynchronous task/job
queue (Celery) and the container-based virtualization platform (Docker). The workflow system
of EcoPAD also provides structured result access and visualization.
**2.2.4.1 Metadata catalog and data management**

Datasets can be placed and queried in EcoPAD via a common metadata catalog which

allows for effective management of diverse data streams. Calls are common for good
management of current large and heterogeneous ecological datasets [*Ellison*, 2010; *Michener*
*and Jones*, 2012; *Vitolo et al.*, 2015]. Kepler [*Ludascher et al.*, 2006] and the Analytic Web
[*Osterweil et al.*, 2010] are two example systems that endeavor to provide efficient data
management through storage of metadata including clear documentation of data
provenance. Similarly to these systems, EcoPAD takes advantage of modern information
technology, especially the metadata catalog, to manage diverse data streams. The EcoPAD
metadata schema includes description of the data product, security, access pattern, and
timestamp of last metadata update *etc*. We use MongDB (https://www.mongodb.com/ ), a
NoSQL database technology, to manage heterogeneous datasets to make the documentation,
query and storage fast and convenient. Through MongDB, measured datasets can be easily fed
into ecological models for various purposes such as to initialize the model, calibrate model
parameters, evaluate model structure and drive model forecast. For datasets from real time
ecological sensors that are constantly updating, EcoPAD is set to automatically fetch new data
streams with adjustable frequency depending on research needs.



### 2.2.4.2 Web API, asynchronous task queue and docker


The RESTful application-programming interface (API) which can deliver data to a wide
variety of applications is the gateway of EcoPAD and enables a wide array of user-interfaces and
data-dissemination activities. Once a user makes a request, such as through clicking on relevant
buttons from a web browser, the request is passed through the Representational State Transfer
(i.e., RESTful) API to trigger specific tasks. The RESTful API bridges the talk between the
client (e.g., a web browser or command line terminal) and the server (Figure 3). The API exploits
the full functionality and flexibility of the HyperText Transfer Protocol (HTTP), such that data
can be retrieved and ingested from the EcoPAD through the use of simple HTTP headers and
verbs (e.g., GET, PUT, POST, *etc*.). Hence, a user can incorporate summary data from EcoPAD
into a website with a single line of html code. Users will also be able to access data directly
through programming environments like R, Python and Matlab. Simplicity, ease of use and
interoperability are among the main advantages of this API which enables web-based modeling.
Celery (https://github.com/celery/celery ) is an asynchronous task/job queue that run at
the background (Figure 3). The task queue (i.e., Celery) is a mechanism used to distribute work
across work units such as threads or machines. Celery communicates through messages, and
EcoPAD takes advantage of the RabbitMQ (https://www.rabbitmq.com/) to manage messaging.
After the user submit a command, the request or message is passed to Celery via the RESTful
API. These messages may trigger different tasks, which include, but not limited to, pull data
from a remote server where original measurements are located, access data through metadata
catalog, run model simulation with user specified parameters, conduct data assimilation which
recursively updates model parameters, forecast future ecosystem status and post-process of
model results for visualization. The broker inside Celery receives task messages and handles out



tasks to available Celery workers which perform the actual tasks (Figure 3). Celery workers are
in charge of receiving messages from the broker, executing tasks and returning task results. The
worker can be a local or remote computation resource (e.g., the cloud) that has connectivity to
the metadata catalog. Workers can be distributed into different IT infrastructures, which makes
EcoPAD workflow easily expandable. Each worker can perform different tasks depending on
tools installed in each worker. And one task can also be distributed into different workers. In
such a way, EcoPAD workflow enables parallelization and distributed computation of actual
modeling tasks across various IT infrastructures, and is flexible in implementing additional
computational resources by connecting additional workers.

Another key feature that makes EcoPAD easily portable and scalable among different

operation systems is the utilization of the container-based virtualization platform, the docker.
Docker can run many applications which rely on different libraries and environments on a single
kernel with its lightweight containerization. Tasks that execute TECO in different ways are
wrapped inside different docker containers that can "talk" with each other. Each docker container
embeds the ecosystem model into a complete filesystem that contains everything needed to run
an ecosystem model: the source code, model input, run time, system tools and libraries. Docker
containers are both hardware-agnostic and platform-agnostic, and they are not confined to a
particular language, framework or packaging system. Docker containers can be run from a
laptop, workstation, virtual machine, or any cloud compute instance. This is done to support the
widely varied number of ecological models running in various languages (e.g., Matlab, Python,
Fortran, C and C++) and environments. In addition to wrap the ecosystem model into a docker
container, software applied in the workflow, such as the Celery, Rabbitmq and MongoDB, are all



lightweight and portable encapsulations through docker containers. Therefore, the entire
EcoPAD is readily portable and applicable in different environments.
**2.2.4.3 Structured result access and visualization**
EcoPAD enables structured result storage, access and visualization to track and analyze
data-model fusion practice. Upon model task completion, the model wrapper code calls a post
processing callback function. This callback function allows for model specific data requirements
to be added to the model result repository. Each task is associated with a unique task ID and
model results are stored within the local repository that can be queried by the unique task ID.
The easy store and query of model results are realized via the MongoDB and RESTful API
(Figure 3). Researchers are authorized to review and download model results and parameters
submitted for each model run through a web accessible URL (link). EcoPAD webpage also
displays a list of historical tasks (with URL) performed by each user. All current and historical
model inputs and outputs are available to download, including the aggregated results produced
for the graphical web applications. In addition, EcoPAD also provides a task report that contains
all-inclusive recap of parameters submitted, task status, and model outputs with links to all data
and graphical results for each task. Such structured result storage and access make sharing,
tracking and referring to modeling studies instant and clear.
**2.3 Scientific functionality**
Scientific functionality of EcoPAD includes web-based model simulation, estimating
model parameters or state variables, quantifying uncertainty of estimated parameters and
projected states of ecosystems, evaluating model structures, assessing sampling strategies,
conducting ecological forecasting. Those functions can be organized to answer various scientific





questions. In addition to the general description in this section, the scientific functionality of
EcoPAD is also illustrated through a few case studies in the following sections.

EcoPAD is designed to perform web-based model simulation, which greatly reduces the

workload of traditional model simulation through manual code compilation and execution. This
functionality opens various new opportunities for modelers, experimenters and the general
public. Model simulation and result analysis are automatically triggered after a simple click on
the web-embedded button (Appendices Figures A2, A3 A6). Users are freed from repeatedly
compiling code, running code and writing programs to analyze and display model results. Such
ease of use has great potential to popularize complex modeling studies that are difficult or
inaccessible for experimenters and the general public. As illustrated through the outreach
activities from the TreeWatch.Net [*Steppe et al.*, 2016], the potential functionality of such web-
based model simulation goes beyond its scientific value as its societal and educational impacts
are critical in solving ecological issues. The web based model simulation also frees users from
model running environment, platform and software. Users can conduct model simulation and do
analysis as long as they have internet access. For example, ecologists can conduct model
simulation and diagnose the underlying reasons for a sudden increase in methane fluxes while
they are making measurements in the field. Youngsters can study ecological dynamics through
their phones or tablets while they are waiting for the bus. Resource managers can make timely
assessment of different resource utilization strategies on spot of a meeting.

EcoPAD is backed up by data assimilation techniques, which facilitate inference of

model parameters and states based on observations. Ecology have witnessed a growing number
of studies focusing on parameter estimation using inverse modeling or data assimilation as large
volumes of ecological measurements become available. To satisfy the growing need of model



parameterization through observations, EcoPAD streamlines parameter estimations and updates.
Researchers can easily review and download files that record parameter values from EcoPAD
result repository. Since these parameters may have different scientific values, the functionality of
EcoPAD related to parameter estimations can potentially embrace diverse subareas in ecology.
For example, soil scientists can study the acclimation of soil respiration to manipulative warming
through shifts in the distribution of the decomposition rate parameter from EcoPAD. The
threshold parameter beyond which further harvesting of fish might cause a crash of fish stocks
can be easily extracted through fish stock assessment models and observations if mounted to
EcoPAD.
EcoPAD promotes uncertainty analysis, model structure evaluation and error
identification. One of the advantages of the Bayesian statistics is its capacity in uncertainty
analysis compared to other optimization techniques [*Xu et al.*, 2006; *Wang et al.*, 2009; *Zhou et*
*al.*, 2012]. Bayesian data assimilation (e.g., MCMC) takes into account observation uncertainties
(errors), generates distributions of model parameters and enables tracking of prediction
uncertainties from different sources. Uncertainty analysis through data assimilation applied to
areas such as ecosystem phenology, fish life cycle and species migration [*Clark et al.*, 2003;
*Cook et al.*, 2005; *Crozier et al.*, 2008; *Luo et al.*, 2011b], can potentially take advantage of
EcoPAD platform to provide critical information for well informed decisions in face of pressing
global change challenges. In addition, the archive capacity of EcoPAD facilitates inter-
comparisons among different models or different versions of the same model to evaluate model
structures and to disentangle structure uncertainties and errors.
The realization of both the near time and long term ecological forecast is one of the key
innovations of EcoPAD. Forecasting capability of EcoPAD is supported by process based



ecological models, multiple observational or experimental data, inverse parameter estimation and
uncertainty quantification through data assimilation, and forward simulation under future
external conditions. The systematically constrained forecast from EcoPAD is accompanied by
uncertainty/confidence estimates to quantify the amount of information that can actually be
utilized from a study. The automated near time forecast, which is constantly adjusted once new
observational data streams are available, provides experimenters advanced and timely
information to assess and adjust experimental plans. For example, with forecasted and displayed
biophysical and biochemical variables, experimenters could know in advance what the most
likely biophysical conditions are. Knowing if the water table may suddenly go aboveground in
response to a high rainfall forecast in the coming week, could allow researcher to emphasize
measurements associated with methane flux. In such a way, experimenters can not only rely on
historical ecosystem dynamics, but also refer to future predictions. Experimenters will benefit
especially from variables that are difficult to track in field due to situations such as harsh
environment, shortage in man power or on instrument limitation.

Equally important, EcoPAD creates new avenues to answer classic and novel ecological

questions, for example, the frequently reported acclimation phenomena in ecology. While
growing evidence points to altered ecological functions as organisms adjust to the rapidly
changing world [*Medlyn et al.*, 1999; *Luo et al.*, 2001; *Wallenstein and Hall*, 2012], traditional
ecological models treat ecological processes less dynamical, as the governing biological
parameters or mechanisms fails to explain such biological shifts. EcoPAD facilitates the shift of
research paradigm from a fixed process representation to a more dynamic description of
ecological mechanisms with constantly updated and archived parameters constrained by
observations under different conditions. Specifically to acclimation, EcoPAD promotes



quantitatively evaluations while previous studies remain mostly qualitative [*Wallenstein and*
*Hall*, 2012; *Shi et al.*, 2015]. We will further illustrate how EcoPAD can be used to address
different ecological questions in the case studies of the SPRUCE project.

**3 EcoPAD performance at testbed - SPRUCE**
**3.1 SPRUCE project overview**

EcoPAD is being applied to the Spruce and Peatland Responses Under Climatic and

Environmental change (SPRUCE) experiment located at the USDA Forest Service Marcell
Experimental Forest (MEF, 47°30.476' N, 93°27.162' W) in northern Minnesota [*Kolka et al.*,
2011]. SPRUCE is an ongoing project focuses on responses of northern peatland to climate
warming and increased atmospheric $CO_2$ concentration [*Hanson et al.*, 2017]. At SPRUCE
ecologists measure various aspects of responses of organisms (from microbes to trees) and
ecological functions (carbon, nutrient and water cycles) to a warming climate. One of the key
features of the SPRUCE experiments is the manipulative deep soil/peat heating (0-3 m) and
whole ecosystem warming treatments (peat + air warmings) which include tall trees (> 4 m)
[*Hanson et al.*, 2017]. Together with elevated atmospheric $CO_2$ treatments, SPRUCE provides a
platform for exploring mechanisms controlling the vulnerability of organisms, biogeochemical
processes and ecosystems in response to future novel climatic conditions. The SPRUCE peatland
is especially sensitive to future climate change and also plays an important role in feeding back
to future climate change through greenhouse gas emissions as it stores a large amount of soil
organic carbon. Vegetation in the SPRUCE site is dominated by *Picea mariana* (black spruce)
and *Sphagnum spp* (peat moss). The studied peatland also has an understory which include
ericaceous and woody shrubs. There are also a limited number of herbaceous species. The whole





ecosystem warming treatments include a large range of both aboveground and belowground
temperature manipulations (ambient, control plots of + 0 °C, +2.25 °C, +4.5 °C, +6.75 °C and +9
°C) in large 115 m$^2$ open-topped enclosures with elevated $CO_2$ manipulations (+0 or +500 ppm).
The SPRUCE project generates a large variety of observational datasets that reflect
ecosystem dynamics from different scales and are available from the project webpage
(https://mnspruce.ornl.gov/) and FTP site (ftp://sprucedata.ornl.gov/). These datasets come from
multiple sources: half hourly automated sensor records, species surveys, laboratory
measurements, laser scanning images *etc*. Involvements of both modeling and experimental
studies in the SPRUCE project create the opportunity for data-model communication. Datasets
are pulled from SPRUCE archives and stored in the EcoPAD metadata catalog for running the
TECO model, conducting data-model fusion or forecasting. The TECO model has been applied
to simulate and forecast carbon dynamics with productions of $CO_2$ and $CH_4$ from different
carbon pools, soil temperature response, snow depth and freeze-thaw cycles at the SRPUCE site
[*Huang et al.*, 2017; *Ma et al.*, 2017; *Jiang et al.*, 2018].

**3.2 EcoPAD-SPRUCE web portal**
We assimilate multiple streams of data from the SPRUCE experiment to the TECO
model using the MCMC algorithm, and forecast ecosystem dynamics in both near time and for
the next 10 years. Our forecasting system for SPRUCE is available at
https://ecolab.nau.edu/ecopad_portal/. From the web portal, users can check our current near and
long term forecasting results, conduct model simulation, data assimilation and forecasting runs,
and analyze/visualize model results. Detailed information about the interactive web portal is
provided in the Appendices.





### 3.3 Near time ecosystem forecasting and feedback to experimenters


As part of the forecasting functionality, EcoPAD-SPRUCE automates the near time
(weekly) forecasting with continuously updated observations from SPRUCE experiments (Figure
5). We set up the system to automatically pull new data streams every Sunday from the SPRUCE
FTP site that holds observational data and update the forecasting results based on new data
streams. Updated forecasting results for the next week are customized for the SPRUCE
experiments with different manipulative treatments and displayed in the EcoPAD-SPRUCE
portal. At the same time, these results are sent back to SPRUCE communities and displayed
together with near term observations for experimenter's reference.
**3.4 New approaches to ecological studies towards better forecasting**
**3.4.1 Case 1: Interactive communications among modelers and experimenters**
EcoPAD-SPRUCE provides a platform to stimulate interactive communications between
modelers and experimenters. Models require experimental data to constrain initial conditions and
parameters, and to verify model performance. A reasonable model is built upon correct
interpretation of information served by experimenters. Model simulations on the other hand can
expand hypotheses testing, and provide thorough or advanced information to improve field
experiments. Through recursively exchanging information between modelers and experimenters,
both models and field experiments can be improved. As illustrated in Figure 5, through extensive
communication between modelers and experimenters, modelers generate model predictions.
Model predictions provide experimenters advanced information, help experimenters think,
question and understand their experiments. Questions raised by experimenters stimulate further
discussion and communication. Through communication, models or/and measurements are
adjusted. With new measurements or/and strengthened models, a second round of prediction is



highly likely to be improved. As the loop of prediction-question-discussion-adjustment-
prediction goes on, forecasting is informed with best understandings from both data and model.

We illustrate how the prediction-question-discussion-adjustment-prediction cycle and

stimulation of modeler-experimenter communication improves ecological predictions through
one episode during the study of the relative contribution of different pathways to methane
emissions. An initial methane model was built upon information (e.g., site characteristics and
environmental conditions) provided by SPRUCE field scientists, taking into account important
processes in methane dynamics, such as production, oxidation and emissions through three
pathways (i.e., diffusion, ebullition and plant-mediated transportation). The model was used to
predict relative contributions of different pathways to overall methane emissions under different
warming treatments after being constrained by measured surface methane fluxes. Initial
forecasting results which indicated a strong contribution from ebullition under high warming
treatments were sent back to the SPRUCE group. Experimenters doubted about such a high
contribution from the ebullition pathway and a discussion was stimulated. It is difficult to
accurately distinguish the three pathways from field measurements. Field experimenters
provided potential avenues to extract measurement information related to these pathways, while
modelers examined model structure and parameters that may not be well constrained by
available field information. Detailed discussion is provided in Table 1. After extensive
discussion, several adjustments were adopted as a first step to move forward. For example, the
three-porosity model that was used to simulate the diffusion process was replaced by the
Millington-Quirk model to more realistically represent methane diffusions in peat soil; the
measured static chamber methane fluxes were also questioned and scrutinized more carefully to
clarify that they did not capture the episodic ebullition events. Measurements such as these





related to pore water gas data may provide additional inference related to ebullition. The updated
forecasting is more reasonable than the initial results although more studies are in need to
ultimately quantify methane fluxes from different pathways.
**3.4.2 Case 2: Acclimation of ecosystem carbon cycling to experimental manipulations**
As a first step, $CH_4$ static chamber flux measurements were assimilated into TECO to
assess potential acclimation phenomena during methane production under 5 warming treatments
(+0, +2.25, +4.5, +6.75, +9 °C). Initial results indicated a reduction in both the $CH_4:CO_2$ ratio
and the temperature sensitivity of methane production based on their posterior distributions
(Figure 6). The mean $CH_4:CO_2$ ratio decreased from 0.675 (control) to 0.505 (+9 °C treatment),
while the temperature sensitivity ($Q_{10}$) for $CH_4$ production decreased from 3.33 (control) to 1.22
(+9 °C treatment). Such shifts quantify potential acclimation of methane production to warming
and future climate warming is likely to have a smaller impact on emission than most of current
predictions that do not take into account of acclimation.
Despite these results are preliminary as more relevant datasets are under collection with
current ongoing warming manipulation and measurements, assimilating observations through
EcoPAD provides a quantitative approach to timely assess acclimation through time. *Melillo et*
*al.* [2017] revealed that the thermal acclimation of the soil respiration in the Harvard Forest is
likely to be phase (time) dependent during their 26-year soil warming experiment. EcoPAD
provides the possibility in tracing the temporal path of acclimation with its streamlined structure
and archive capacity. *Shi et al.* [2015] assimilated carbon related measurements in a tallgrass
prairie into the TECO model to study acclimation after 9-years warming treatments. They
revealed a reduction in the allocation of GPP to shoot, the turnover rates of the shoot and root
carbon pools, and an increase in litter and fast carbon turnovers in response to warming





treatments. Similarly, as time goes on, the SPRUCE experiment will generate more carbon
cycling related datasets under different warming and $CO_2$ treatments, which can be mounted to
EcoPAD to systematically quantify acclimations in carbon cycling.
**3.4.3 Case 3: Partitioning of uncertainty sources**

Uncertainties in ecological studies can come from observations (include forcing that

drives the model), different model structures to represent the real world and the specified model
parameters [*Luo et al.*, 2016]. Previous studies tended to focus on one aspect of the uncertainty
sources instead of disentangling the contribution from different sources. For example, the model
intercomparison projects (MIPs), such as TRENDY, focus on uncertainty caused by different
model structures with prescribed external forcing [*Sitch et al.*, 2008]. *Keenan et al.* [2012] used
data assimilation to constrain parameter uncertainties in projecting Harvard forest carbon
dynamics. *Ahlstrom et al.* [2012] forced one particular vegetation model by 18 sets of forcings
from climate models of the Coupled Model Intercomparison Project Phase 5 (CMIP5), while the
parameter or model structure uncertainty is not taken into account.

EcoPAD is designed to provide a thorough picture of uncertainties from multiple sources

especially in carbon cycling studies. Through focusing on multiple instead of one source of
uncertainty, ecologists can allocate resources to areas that cause relative high uncertainty.
Attribution of uncertainties in EcoPAD relies on an ensemble of ecosystem models, the data
assimilation system and climate forcing with quantified uncertainty. For example, *Jiang et al.*
[2018] focused specifically on the relative contribution of parameter uncertainty vs. climate
forcing uncertainty in forecasting carbon dynamics at the SPRUCE site. Through assimilating
the pre-treatment measurements (2011-2014) from the SPRUCE experiment, *Jiang et al.* [2018]
estimated uncertainties of key parameters that regulate the peatland carbon dynamics. Combined



with the stochastically generated climate forcing (e.g., precipitation and temperature), *Jiang et al.*
[2018] found external forcing resulted in higher uncertainty than parameters in forecasting
carbon fluxes, but caused lower uncertainty than parameters in forecasting carbon pools.
Therefore, more efforts are required to improve forcing measurements for studies that focus on
carbon fluxes (e.g., GPP), while reductions in parameter uncertainties are more important for
studies in carbon pool dynamics. Such kind of uncertainty assessment benefits from EcoPAD
with its systematically archived model simulation, data assimilation and forecasting.
**3.4.4 Case 4: Improving biophysical estimation for better ecological prediction**

Carbon cycling studies can also benefit from EcoPAD through improvements in external

forcing. Soil environmental condition is an important regulator of belowground biological
activities and also feeds back to aboveground vegetation growth. Biophysical variables such as
soil temperature, soil moisture, ice content and snow depth, are key predictors of ecosystem
dynamics. After constraining the biophysical module by detailed monitoring data from the
SPRUCE experiment through the data assimilation component of EcoPAD, *Huang et al.* [2017]
forecasted the soil thermal dynamics under future conditions and studied the responses of soil
temperature to hypothetical air warming. This study emphasized the importance of accurate
climate forcing in providing robust thermal forecast. In addition, *Huang et al.* [2017] revealed
non-uniform responses of soil temperature to air warming. Soil temperature responded stronger
to air warming during summer compared to winter. And soil temperature increased more in
shallow soil layers compared to deep soils in summer in response to air warming. Therefore,
extrapolating of manipulative experiments based on air warming alone may not reflect the real
temperature sensitivity of SOM if soil temperature is not monitored. As robust quantification of
environmental conditions is known to be a first step towards better understanding of ecological





process, improvement in soil thermal predictions through EcoPAD data assimilation system is
helpful in telling apart biogeochemical responses from environmental uncertainties and also in
providing field ecologists beforehand key environmental conditions.
**3.4.5 Case 5: How do updated model and data contribute to reliable forecasting?**

Through constantly adjusted model and external forcing according to observations and

weekly archived model parameter, model structure, external forcing and forecasting results, the
contribution of model and data updates can therefore be tracked through comparing forecasted vs.
realized simulations. For example, Figure 7 illustrates how realized external forcing (compared
to stochastically generated forcing) and shifts in ecosystem state variables shape ecological
predictions. Similarly as in other EcoPAD-SPURCE case studies, TECO is trained through data
assimilation with observations from 2011-2014 and is used to forecast GPP and total soil organic
carbon content at the beginning of 2015. For demonstrating purpose, Figure 7 only shows 3
series of forecasting results instead of updates from every week. Series 1 (S1) records forecasted
GPP and soil carbon with stochastically generated weather forcing from January 2015-December
2024 (Figure 7a,b cyan). Series 2 (S2) records simulated GPP and soil carbon with observed
climate forcing from January 2015 to July 2016 and forecasted GPP and soil carbon with
stochastically generated forcing from August 2016 - December 2024 (Figure 7a,b red). Similarly,
the stochastically generated forcing in Series 3 (S3) starts from January 2017 (Figure 7a,b blue).
For each series, predictions were conducted with randomly sampled parameters from the
posterior distributions and stochastically generated forcing. We displayed 100 mean values
(across an ensemble of forecasts with different parameters) corresponding to 100 forecasts with
stochastically generated forcing.





GPP is highly sensitive to climate forcing. The differences between the realized (S2, 3)
and initial forecasts (S1) reach almost 800 gC m$^{-2}$ year$^{-1}$ (Figure 7c). The discrepancy is strongly
dampened in the following 1-2 years. The impact of realized forecasts is close to 0 after
approximately 5 years. However, soil carbon pool shows a different pattern. Soil carbon pool is
increased by less than 150 gC m$^{-2}$, which is relative small compared to the carbon pool size of *ca.*
62000 gC m$^{-2}$. The impact of realized forecasts grows with time and reaches the highest at the
end of the simulation year 2024. GPP is sensitive to the immediate change in climate forcing
while the updated ecosystem status (or initial value) has minimum impact in the long term
forecast of GPP. The impact of updated climate forcing is relatively small for soil carbon
forecasts during our study period. Soil carbon is less sensitive to the immediate change of
climate compared to GPP. However, the alteration of system status affects soil carbon forecast
especially in a longer time scale.
Since we are archiving realized forecasts every week, we can track the relative
contribution of ecosystem status, forcing uncertainty and parameter distributions to the overall
forecasting patterns of different ecological variables and how these patterns evolve in time. In
addition, as growing observations of ecological variables (e.g., carbon fluxes and pool sizes)
become available, it is feasible to diagnose key factors that promote robust ecological forecasting
through comparing the archived forecasts vs. observation and analysing archives of model
parameters, initial values and climate forcing *etc*.

**4 Discussion**
**4.1 The necessity of interactive infrastructure to realize ecological forecasting**



Substantial increases in data availability from observational and experimental networks,
surges in computational capability, advancements in ecological models and sophisticated
statistical methodologies and pressing societal need for best management of natural resources
have shifted ecology to emphasis more on quantitative forecasts. However, quantitative
ecological forecast is still young and our knowledge about ecological forecasting is relatively
sparse, inconsistent and disconnected [*Luo et al.*, 2011b; *Petchey et al.*, 2015]. Therefore, both
optimistic and pessimistic viewpoints exist on the predictability of ecology [*Clark et al.*, 2001;
*Beckage et al.*, 2011; *Purves et al.*, 2013; *Petchey et al.*, 2015; *Schindler and Hilborn*, 2015].
Ecological forecasting is complex and advantages in one single direction, for example,
observations alone or statistical methodology alone, is less likely to lead to successful forecasting
compared to approaches that effectively integrate improvements from multiple sectors.
Unfortunately, realized ecological forecasting that integrates available resources is relative rare
due to lack of relevant infrastructures.
EcoPAD provides such effective infrastructure with its interactive platform that
rigorously integrates merits from models, observations, statistical advance, information
technology and human resources from experimenter, modeler as well as the general public to
best inform ecological forecasting, boost forecasting practice and delivery of forecasting results.
Interactions enable exchanging and extending of information so as to benefit from collective
knowledge. For example, manipulative studies will have a much broader impact if the
implications of their results can be extended from the regression between environmental variable
and ecosystem response, such as be integrated into an ecosystem model through model-data
communication. Such an approach will allow gaining information about the processes
responsible for ecosystem's response, constraining models, and making more reliable



predictions. Going beyond common practice of model-data assimilation from which model
updating lags far behind observations, EcoPAD enables iterative model updating and forecasting
through dynamically integrating models with new observations in near real time. This real-time
interactive capacity relies on its scientific workflow that automates data management, model
simulation, data simulation and result visualization. The open, timely, convenient, transparent,
flexible, reproducible and traceable characteristics of this platform, also thanks to its scientific
workflow, encouraged thorough interactions between experimenters and modelers. Forecasting
results from SPRUCE were timely shared among research groups with different background
through the web interface. Expertise from different research groups was integrated to improve a
second round of forecasting. Again, thanks to the workflow, new information or adjustment is
relatively easy to incorporate into future forecasting, making the forecasting system fully
interactive and dynamical.

We also benefit from the interactive EcoPAD platform to broaden user-model

interactions and to broadcast forecasting results. Learning about the ecosystem models and data-
model fusion techniques may lag one's productivity and even discourage learning the modeling
techniques because of their complexity and long learning curve. Because EcoPAD can be
accessed from a web browser and does not require any coding from the user's side, the time lag
between learning the model structure and obtaining model-based results for one's study is
minimal, which opens the door for non-modeler groups to "talk" with models. The online storage
of one's results lowers the risk of data loss. The results of each model run can be easily tracked
and shared with its unique ID and web address. In addition, the web-based workflow also saves
time for experts with automated model running, data assimilation, forecasting, structured result
access and instantaneous graphic outputs, bringing the possibility for thorough exploration of



more essence part of the system. The simplicity in use of EcoPAD at the same time may limit
their access to the code and lowers the flexibility. Flexibility for users with higher demands, for
example, those who wanted to test alternative data assimilation methods, use a different carbon
cycle model, change the number of calibrated parameters, include the observations for other
variables, is provided through the GitHub repository (https://github.com/ou-ecolab ). This
GitHub repository contains code and instruction for installing, configuring and controlling the
whole system, users can easily adapt the workflow to wrap their own model based on his or her
needs.
In addtion to benefit from its workflow, the advantage of EcoPAD is also reflected in its
data assimilation capacity especially for land carbon studies. One focus of EcoPAD is to
constrain parameters of terrestrial carbon models to predict long-term carbon dynamics (e.g., 100
years) which are determined more by parameters than initial values of state variables [*Weng and*
*Luo*, 2011]. EcoPAD incorporates the Bayesian framework, especially the MCMC method, to
constrain parameters. In comparison, DART uses the Ensemble Kalman Filter to adjust model
state variables, instead of parameters, to match observations over time. In the past, complex
models could not assimilate pool-related data to constrain their parameter estimation due to
insurmountable computational demand in large scale studies. For example, CCDAS normally
only assimilates flux-based data [*Peylin et al.*, 2016]. EcoPAD is flexible in assimilating both
pool- and flux-based data into complex models so that both fluxes and turnover rates of pools
can be constrained with its matrix representation [*Hararuk et al.*, 2014; *Luo*, 2017] and its
capability to wrap different models.
**4.2 Implications for better ecological forecasting**



Specifically to reliable forecasting of carbon dynamics, our initial exploration from
EcoPAD-SPRUCE indicates that realistic model structure, correct parameterization and accurate
external environmental conditions are essential. Model structure captures important known
mechanisms that regulate ecosystem carbon dynamics. Adjustment in model structure is critical
in our improvement in methane forecasting. Model parameters may vary between observation
sites, change with time or environmental conditions [*Medlyn et al.*, 1999; *Luo et al.*, 2001]. A
static or wrong parameterization misses important mechanisms (e.g., acclimation and adaptation)
that regulate future carbon dynamics. Correct parameterization is especially important for long
term carbon pool predictions as parameter uncertainty resulted in high forecasting uncertainty in
our case study [*Jiang et al.*, 2018]. Although the picture about how neglecting of parameter shift
affects carbon predictions has not yet been fully revealed from EcoPAD-SPRUCE as field
measurements are still ongoing, our initial exploration indicates non-negligible acclimation of
ecosystem methane production in response to warming. External environmental condition is
another important factor in carbon predictions. External environmental condition includes both
the external climatic forcing that is used to drive ecosystem models and also the environmental
condition that is simulated by ecosystem models. As we showed that air warming may not
proportionally transfer to soil warming, realistic soil environmental information needs to be
appropriately represented to predict soil carbon dynamics [*Huang et al.*, 2017]. The impact of
external forcing is especially obvious in short term carbon flux predictions. Forcing uncertainty
resulted in higher forecasting uncertainty in carbon flux compared to that from parameter
uncertainty [*Jiang et al.*, 2018]. Mismatches in forecasted vs. realized forcing greatly increased
simulated GPP and the discrepancy diminished in the long run. Reliable external environmental
condition, to some extent, reduces the complexity in diagnosing modeled carbon dynamics.



Pool-based vs. flux-based predictions are regulated differently by external forcing and

initial states, which indicates that differentiated efforts are required to improve short vs. long
term predictions. External forcing, which has not been well emphasized in previous carbon
studies, has strong impact on short term forecasting. The large response of GPP to forecasted vs.
realized forcing as well the stronger forcing-caused uncertainty in GPP predictions indicate
correct forcing information is a key step in short term flux predictions. In this study, we
stochastically generated the climate forcing based on local climatic conditions (1961-2014),
which is not sufficient in capturing local short term climate variability. As a result, realized GPP
went outside our ensemble forecasting. On the other hand, parameters and historical information
about pool status are more important in long term pool predictions. Therefore, improvement in
long term pool size predictions cannot be reached by accurate climatic information alone.
Instead, it requires accumulation in knowledge related to site history and processes that regulate
pool dynamics.

Furthermore, reliable forecasting needs understanding of uncertainty sources in addition

to the future mean states. Uncertainty and complexity are major reasons that lead to the belief in
"computationally irreducible" and low intrinsic predictability of ecological systems [*Coreau et*
*al.*, 2010; *Beckage et al.*, 2011; *Schindler and Hilborn*, 2015]. Recent advance in computational
statistical methods offers a way to formally accounting for various uncertainty sources in
ecology [*Clark et al.*, 2001; *Cressie et al.*, 2009]. And the Bayesian approach embedded in
EcoPAD brings the opportunity to understand and communicate forecasting uncertainty. Our
case study revealed that forcing uncertainty is more important in flux-based predictions while
parameter uncertainty is more critical in pool-based predictions. Actually, how uncertainty in
carbon forecasting changes with time, what are the dominate sources of uncertainty (parameter,



initial condition, model structure, observation errors, forcing *etc.*) under different conditions,
how uncertainty sources interact among different components are all valuable questions that can
be explored through EcoPAD.
**4.3 Applications of EcoPAD to manipulative experiments and observation sites**
Broadly speaking, data-model integration stands to increase the overall precision and
accuracy of model-based experimentation [*Luo et al.*, 2011b; *Niu et al.*, 2014]. Systems for
which data have been collected in the field and which are well represented by ecological models
therefore have the capacity to receive the highest benefits from EcoPAD to improve forecasts. In
a global change context, experimental manipulations including ecosystem responses to changes
in precipitation regimes, carbon dioxide concentrations, temperatures, season lengths, and
species compositional shifts can now be assimilated into ecosystem models [*Xu et al.*, 2006; *Gao*
*et al.*, 2011; *Lebauer et al.*, 2013; *Shi et al.*, 2016]. Impacts of these global change factors on
carbon cycling and ecosystem functioning can now be measured in a scientifically transparent
and verifiable manner. This leads to ecosystem modeling of systems and processes that can
obtain levels of confidence that lend credibility with the public to the science's forward progress
toward forecasting and predicting [*Clark et al.*, 2001]. These are the strengths of a widely-
available interface devoted to data-model integration towards better forecasting.
The data-model integration framework of EcoPAD creates a smart interactive model-
experiment (ModEx) system. ModEx has the capacity to form a feedback loop in which field
experiment guides modeling and modeling influences experimental focus [*Luo et al.*, 2011a]. We
demonstrated how EcoPAD works hand-in-hand between modelers and experimenters in the life-
cycle of the SPRUCE project. Field experiment from SPRUCE community provides basic data to
set up the ecosystem model and update model parameters recursively, while the forecasting from



ecosystem modeling informs experimenters the potential key mechanisms that regulate
ecosystem dynamics and help experimenters to question and understand their measurements. The
EcoPAD-SPRUCE system operates while experimenters are making measurements or planning
for future researches. Information is constantly fed back between modelers and experimenters,
and simultaneous efforts from both parties illustrate how communications between model and
data advance and shape our understanding towards better forecasts during the lifecycle of a
scientific project. ModEx can be easily extended to other experimental systems to: 1, predict
what might be an ecosystem's response to treatments once experimenter selected a site and
decided the experimental plan; 2, assimilate data experimenters are collecting along the
experiment to constrain model predictions; 3, project what an ecosystem's responses may likely
be in the rest of the experiment; 4, tell experimenters what are those important datasets
experimenters may want to collect in order to understand the system; 5, periodically updates the
projections; and 6, improve the models, the data assimilation system, and field experiments
during the process.

In addition to the manipulative experimental, the data assimilation system of EcoPAD

can be used for automated model calibration for FLUXNET sites or other observation networks,
such as the NEON and LTER [*Johnson et al.*, 2010; *Robertson et al.*, 2012]. The application of
EcoPAD at FLUXNET, NEON or LTER sites includes three steps in general. First, build the
climate forcing in the suitable formats of EcoPAD from the database of each site; Second, collect
the prior information (include observations of state variables) in the data assimilation system
from FLUXNET, NEON or LTER sites; Third, incorporate the forcing and prior information into
EcoPAD, and then run the EcoPAD with the dynamic data assimilation system. Furthermore,
facing the proposed continental scale ecology study [*Schimel*, 2011], EcoPAD once properly



applied could also help evaluate and optimize field deployment of environmental sensors and
supporting cyberinfrastructure, that will be necessary for larger, more complex environmental
observing systems being planned in the US and across different continents. Altogether, with its
milestone concept, EcoPAD benefits from observation and modeling and at the same time
advances both observation and modeling of ecological studies.
**4.4 Future developments**
As we indicated, EcoPAD will expand as time goes on. The system is designed to
incorporate multiple biogeochemical models, diverse data assimilation techniques and various
ecosystem state variables. A multiple (or ensemble) model approach is helpful in tracking
uncertainty sources from our process understanding. With rapid evolving ecological knowledge,
emerging models with different hypotheses, such as the microbial-enzyme model [*Wieder et al.*,
2013], enhance our capacity in ecological prediction but can also benefit from rapid tests against
data if incorporated into EcoPAD. In addition to MCMC [*Braswell et al.*, 2005; *Xu et al.*, 2006],
a variety of data assimilation techniques have been recently applied to improve models for
ecological forecasting, such as the EnKF [*Gao et al.*, 2011], Genetic Algorithm [*Zhou and Luo*,
2008] and 4-d variational assimilation [*Peylin et al.*, 2016]. Future development will incorporate
different optimization techniques to offer users the option to search for the best model
parameters by selecting and comparing the possibly best method for their specific study. We
focus mostly on carbon related state variables in the SPRUCE example, and the data assimilation
system in EcoPAD needs to include more observed variables for constraining model parameters.
For example, the NEON sites not only provide measured ecosystem $CO_2$ fluxes and soil carbon
stocks, but also resources (e.g., GPP/Transpiration for water and GPP/intercepted PAR for light)
use efficiency [*Johnson et al.*, 2010]. With these improvements, one goal of the EcoPAD is to





enable the research community to run models and forecast various aspects of future
biogeochemical changes as data becomes available.

The power of EcoPAD not only lies in its scientific values, but also in the potential

service it can bring to the society. Forecasting with carefully quantified uncertainty is helpful in
providing support for natural resource manager and policy maker [*Clark et al.*, 2001]. It is
always difficult to bring the complex mathematical ecosystem models to the general public,
which creates a gap between current scientific advance and public awareness. The web-based
interface from EcoPAD makes modeling as easy as possible without losing the connection to the
mathematics behind the models. It will greatly transform environmental education and encourage
citizen science [*Miller-Rushing et al.*, 2012; *Kobori et al.*, 2016] in ecology and climate change
with future outreach activities to broadcast the EcoPAD platform.
**5 Conclusion**

The fully interactive web-based Ecological Platform for Assimilating Data (EcoPAD)

into models aims to promote data-model integration towards predictive ecology through bringing
the complex ecosystem model and data assimilation techniques easily accessible to different
audience. It is supported by meta-databases of biogeochemical variables, libraries of modules of
process models, toolbox of inversion techniques and easily scalable scientific workflow.
Through these components, it automates data management, model simulation, data assimilation,
ecological forecasting, and result visualization, providing an open, convenient, transparent,
flexible, scalable, traceable and readily portable platform to systematically conduct data-model
integration towards better ecological forecasting.

We illustrated several of its functionalities through the Spruce and Peatland Responses

Under Climatic and Environmental change (SPRUCE) experiment. The iterative forecasting



approach from EcoPAD-SPRUCE through the prediction-question-discussion-adjustment-
prediction cycle and extensive communication between model and data creates a new paradigm
to best inform forecasting. In addition to forecasting, EcoPAD enables interactive web-based
approach to conduct model simulation, estimate model parameters or state variables, quantify
uncertainty of estimated parameters and projected states of ecosystems, evaluate model
structures, and assess sampling strategies. Altogether, EcoPAD-SPRUCE creates a smart
interactive model-experiment (ModEx) system from which experimenters can know what an
ecosystem's response might be at the beginning of their experiments, constrain models through
collected measurements, predict ecosystem's response in the rest of the experiments, adjust
measurements to better understand their system, periodically update projections and improve
models, the data assimilation system, and field experiments.

Specifically to forecasting carbon dynamics, EcoPAD-SPRUCE revealed that better

forecasting relies on improvements in model structure, parameterization and accurate external
forcing. Accurate external forcing is critical for short-term flux-based carbon predictions while
right process understanding, parameterization and historical information are essential for long
term pool based predictions. In addition, EcoPAD provides an avenue to disentangle different
sources of uncertainties in carbon cycling studies and to provide reliable forecasts with
accountable uncertainties.

**Code availability:**
EcoPAD portal is available at https://ecolab.nau.edu/ecopad_portal/ and code is provided at the
GitHub repository (https://github.com/ou-ecolab ).
**Data availability:**





Relevant data for this manuscript is available at the SPRUCE project webpage
(https://mnspruce.ornl.gov/ ) and the EcoPAD web portal (https://ecolab.nau.edu/ecopad_portal/
). Additional data can be requested from the corresponding author.
**Competing interests:**
The authors declare that they have no conflict of interest.
**Acknowledgement**:
SPRUCE components of this work (PJH, DMR) are based upon work supported by the U.S.
Department of Energy, Office of Science, Office of Biological and Environmental Research. Oak
Ridge National Laboratory is managed by UT-Battelle, LLC, for the U.S. Department of Energy
under contract DE-AC05-00OR22725.

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





**Tables**
Table 1. Discussion stimulated by EcoPAD-SPRUCE forecasting among modelers and
experimenters on how to improve predictions of the relative contribution of different pathways
of methane emissions

| | Discussion |
|---|---|
| 1 | No strong bubbles are noted at field and a non-observation constrained modeling study at a similar site from another project concluded minor ebullition contribution, which are at odds with TECO result. |
| 2 | $CH_4:CO_2$ ratio might explain the discrepancy. The other modeling study assumed that decomposed C is mainly turned into $CO_2$ and a smaller fraction is turned into $CH_4$. The large $CH_4:CO_2$ ratio at this site may result in higher CH4 flux. It seems that the most "flexible" term is ebullition because any "excess" (above saturation) CH4 is immediately released to ebullition, while the plant transport term is constrained by vegetation data. |
| 3 | Experimental researches on the relative contribution to methane emission from three different pathways are rare. |
| 4 | Current available observations include net surface flux of methane from the large collars, incubation data that should represent methane sources within the profile, and gas/DOC profile data that can indicate active zones within the peat profile. What are additional data needed to constrain relative contribution of different pathways? |
| 5 | I had always thought that peatlands don't bubble much, but the super-sensitive GPS measurements found movements of the surface of the GLAP peatlands consistent with degassing events, and subsurface radar images did show layers that were interpreted as bubble-layers. |
| 6 | Pore water gas data, perhaps $N_2$ or Ar may shed some light on the relative importance of ebullition. |
| 7 | It is really hard to accurately distinguish the three pathways. It has to rely on multiple approaches. Particularly for the SPRUCE site, the vegetation cover varies, vegetation species varies. How many channels each species has affect the transport? Meanwhile, the presence of plant (even not vascular plant) will lead to more gas transport, but as bubbles, rather than plant-mediated transport. |
| 8 | It depends on model structure and algorithm to simulate diffusion, vascular, and ebullition. Most models assume a threshold to allow ebullition. Diffusion is treated in similar ways as ebullition in some models (most one layer or two layers models). For the multiple layers models, the diffusion occurs from bottom to top mm by mm, layer by layer, therefore, the gas diffusion from top layer to atmosphere is considered the diffusion flux. If that is the case, the time step and wind speed and pressure matter (most models do not consider wind and pressure impacts). Plant transport is really dependent on the parameter for plant species, aerenchyma, etc. The gas transportability of plant is associated with biomass, NPP, or root biomass, seasonality of plant growth, etc. in models. All these differences might cause biases in the final flux. |
| 9 | With only the $CH_4$ emission data cannot constrain the relative contribution of three pathways. Concentration data in different soil layers may help constrain. |
| 10 | Diffusion coefficient calculation in TECO adopts the "three-porosity-model" which is ideal for mineral soil, but may not fit the organic soil. "Millington-Quirk model" for should be a better choice for peat soil. |
| 11 | The boundary condition should be taken care of, but it brings in more uncertainties including the wind speed and piston velocity, etc., |
| 12 | $CH_4$ emissions captured in static chambers does not include the episodic ebullition events. So (1) the static chambers underestimate the total methane emission and (2) might need to exclude the ebullition pathway when using the observation data to constrain the $CH_4$ emission. But this point seems haven't been paid attention to in other models. |




**Figure Legends**

**Figure 1** Schema of approaches to forecast future ecological responses from common practice

(the upper panel) and the Ecological Platform for Assimilation of Data (EcoPAD) (bottom

panel). The common practice makes use of observations to develop or calibrate models to make

predictions while the EcoPAD approach advances the common practice through its fully

interactive platform. EcoPAD consists of four major components: experiment/data, model, data

assimilation and the scientific workflow. Data and model are iteratively integrated through its

data assimilation systems to improve forecasting. And its near-real time forecasting results are

shared among research groups through its web interface to guide new data collections. The

scientific workflow enables web-based data transfer from sensors, model simulation, data

assimilation, forecasting, result analysis, visualization and reporting, encouraging broad user-

model interactions especially for the experimenters and the general public with limited

background in modeling.

**Figure 2** The data assimilation system inside the Ecological Platform for Assimilation of Data

(EcoPAD) towards better forecasting of terrestrial carbon dynamics

**Figure 3** The scientific workflow of EcoPAD. The workflow wraps ecological models and data

assimilation algorithms with the docker containerization platform. Users trigger different tasks

through the Representational State Transfer (i.e., RESTful) application-programming interface

(API). Tasks are managed through the asynchronous task queue, Celery. Tasks can be executed

concurrently on a single or more worker servers across different scalable IT infrastructures.

MongoDB is a database software that takes charge of data management in EcoPAD and

RabbitMQ is a message broker.



**Figure 4.** Near time forecasting of EcoPAD-SPRUCE. EcoPAD automatically synchronizes real
time observations from environmental sensors managed by the SPRUCE experimental
communities. Data from observations are assimilated and used to update forecasting. Weekly
forecasting results are displayed in the EcoPAD-SPRUCE web portal
(http://ecolab.cybercommons.org/ecopad_portal/) as well as sent back to the experimental groups
to guide future experimental design and sampling.
**Figure 5.** Schema of interactive communication between modelers and experimenters through
the prediction-question-discussion-adjustment-prediction cycle to improve ecological
forecasting. The schema is inspired by an episode of experimenter-modeler communication
stimulated by the EcoPAD-SPRUCE platform. The initial methane model constrained by static
chamber methane measurements was used to predict relative contributions of three methane
emission pathways (i.e., ebullition, plant mediated transportation (PMT) and diffusion) to the
overall methane fluxes under different warming treatments (+ 0 °C, +2.25 °C, +4.5 °C, +6.75 °C
and +9 °C). The initial results indicated a dominant contribution from ebullition especially under
+9 °C which was doubted by experimenters. The discrepancy stimulated communications
between modelers and experimenters with detailed information listed in Table 1. After extensive
discussion, the model structure was adjusted and field observations were reevaluated. And a
second round of forecasting yielded more reliable predictions.
**Figure 6.** Posterior distribution of the ratio of $CH_4:CO_2$ (panel a) and the temperature sensitivity
of methane production ($Q_{10\_CH4}$, panel b) under 5 warming treatments.
**Figure 7.** Realized vs. unrealized forecasting of gross primary production (panels a,c) and soil
organic C content (panels b,d). The upper panels show 3 series of forecasting with different
weather forcing. Cyan indicates forecasting with 100 stochastically generated weather forcing





from January 2015 to December 2024 (S1); red corresponds to realized forecasting with
measured weather forcing from January 2015 to July 2016 followed by forecasting with 100
stochastically generated weather forcing (S2); and blue shows realized forecasting with
measured weather forcing from January 2015 to December 2016 followed by forecasting with
100 stochastically generated weather forcing (S3). The bottom panels display mismatches
between realized forecasting (S2,3) and the original unrealized forecasting (S1). Red displays the
difference between S2 and S1 (S2-S1) and blue shows discrepancy between S3 and S1 (S3-S1).
Dashed green lines indicates the start of forecasting with stochastically generated weather
forcing. Note that the left 2 panels are plotted on yearly time-scale and the right 2 panels show
results on monthly time-scale.







**Figure 1**

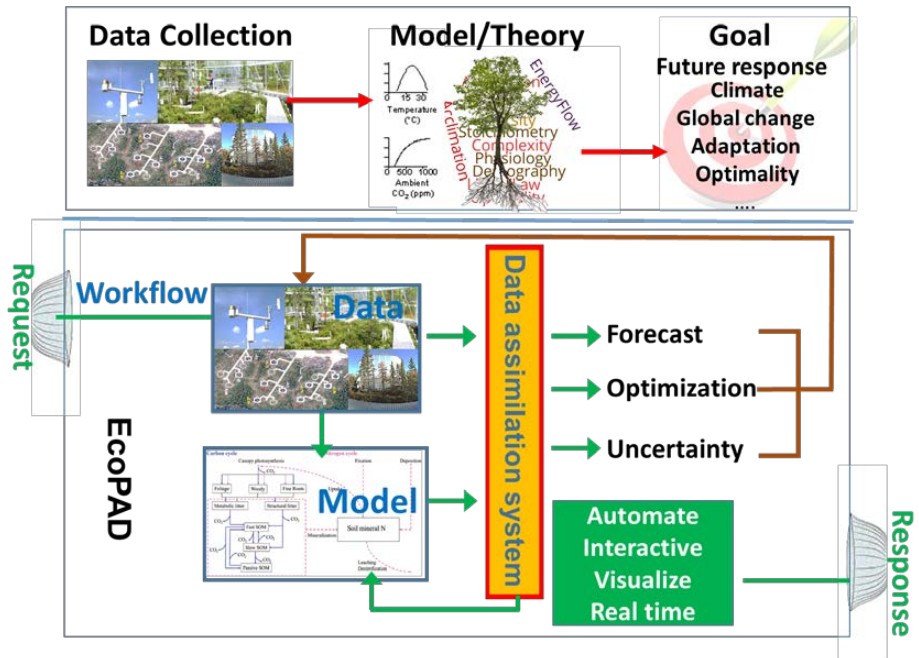







**Figure 2**


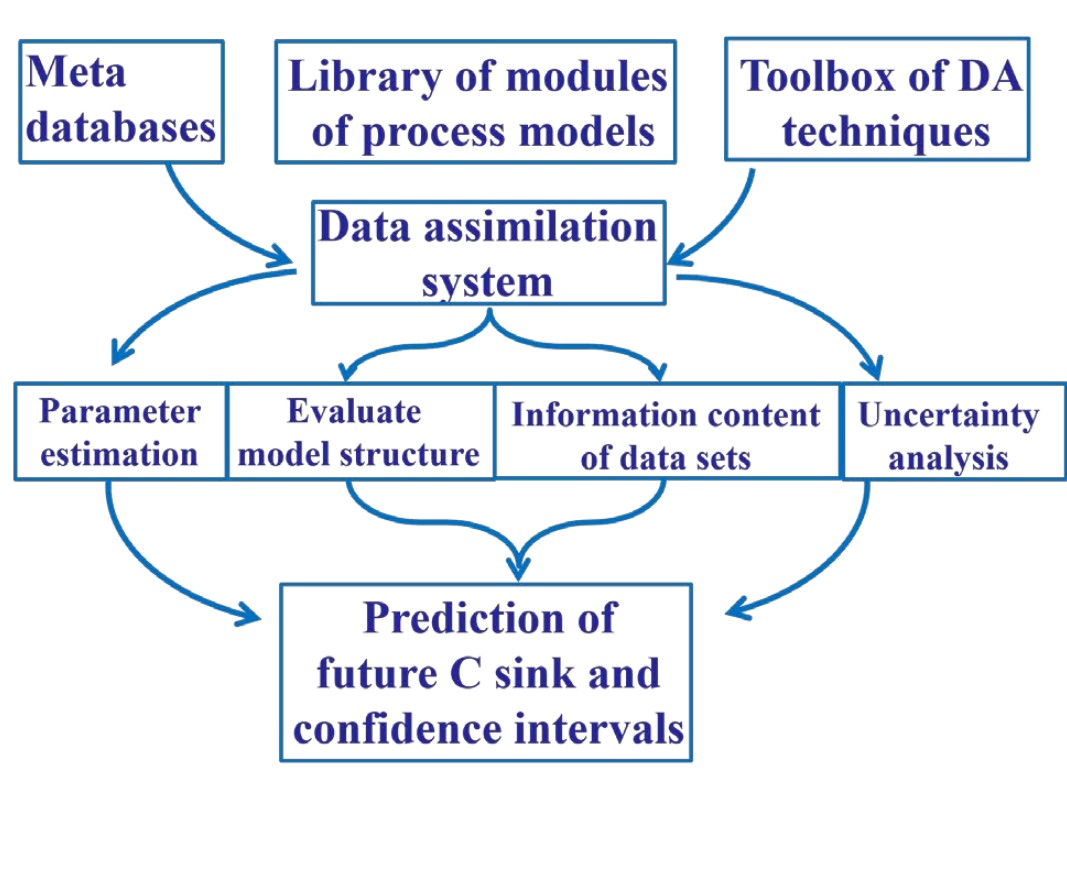










**Figure 3**

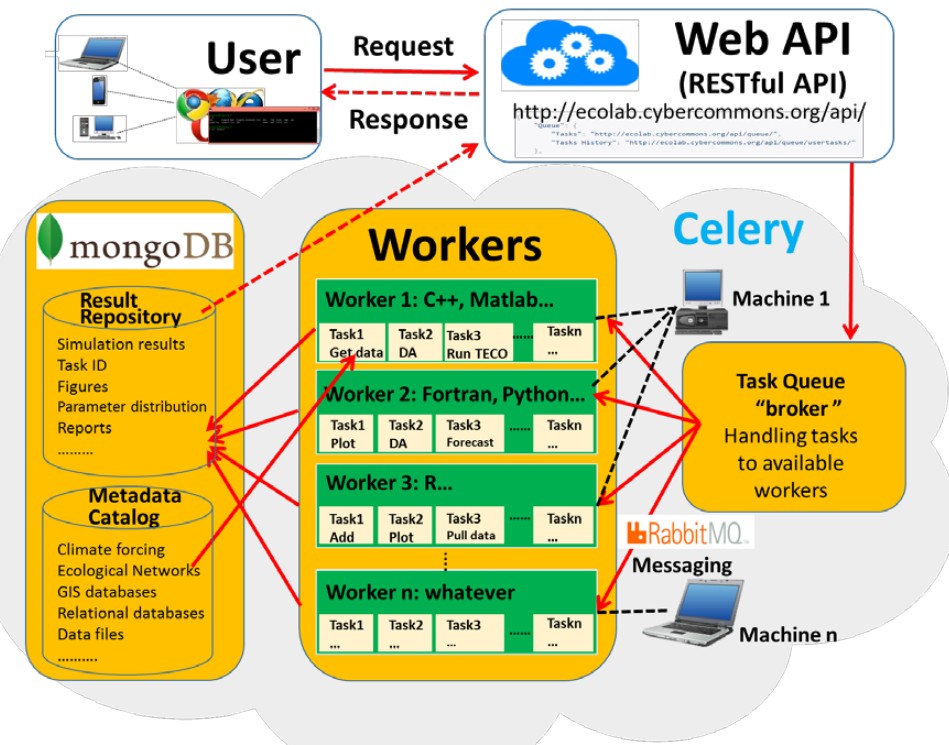






**Figure 4**

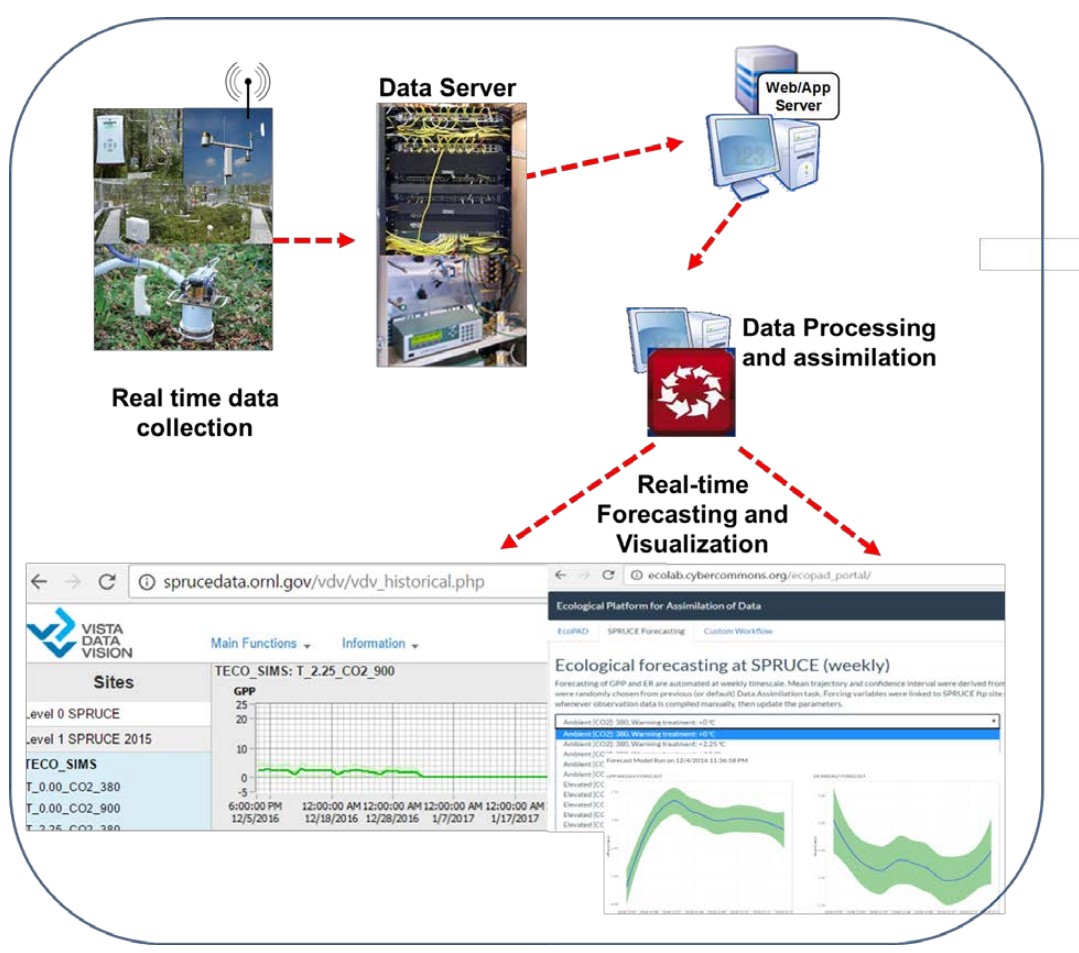







**Figure 5**

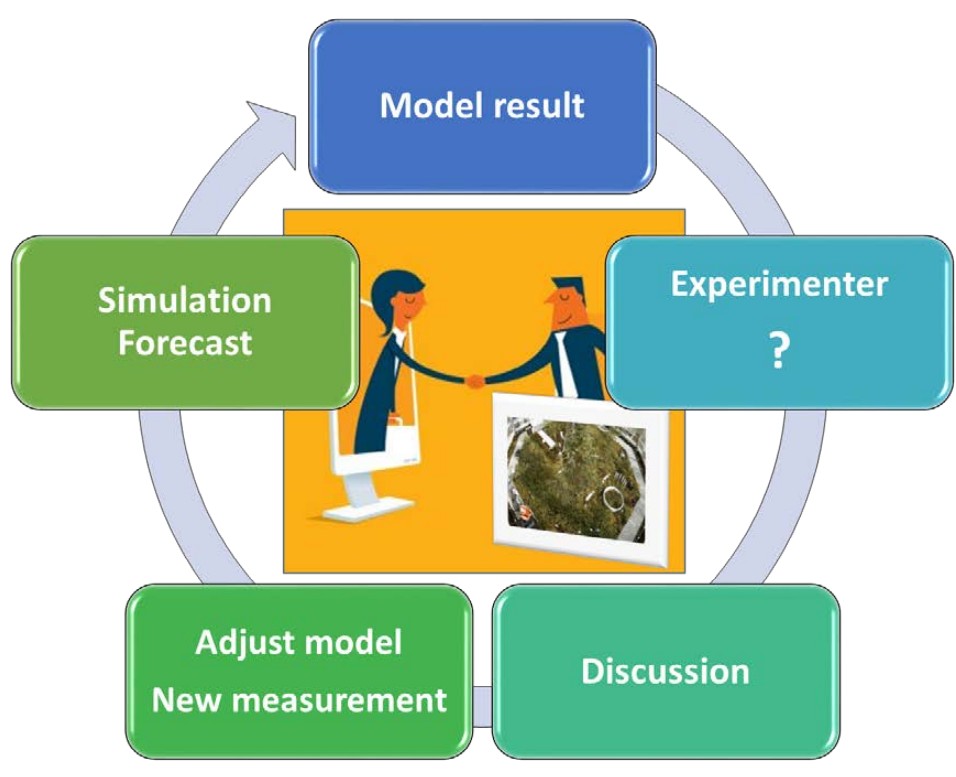






**Figure 6**

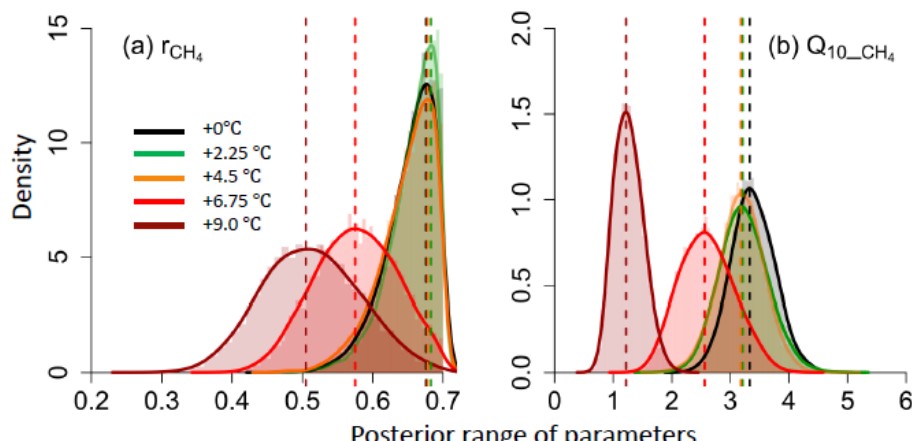








**Figure 7**

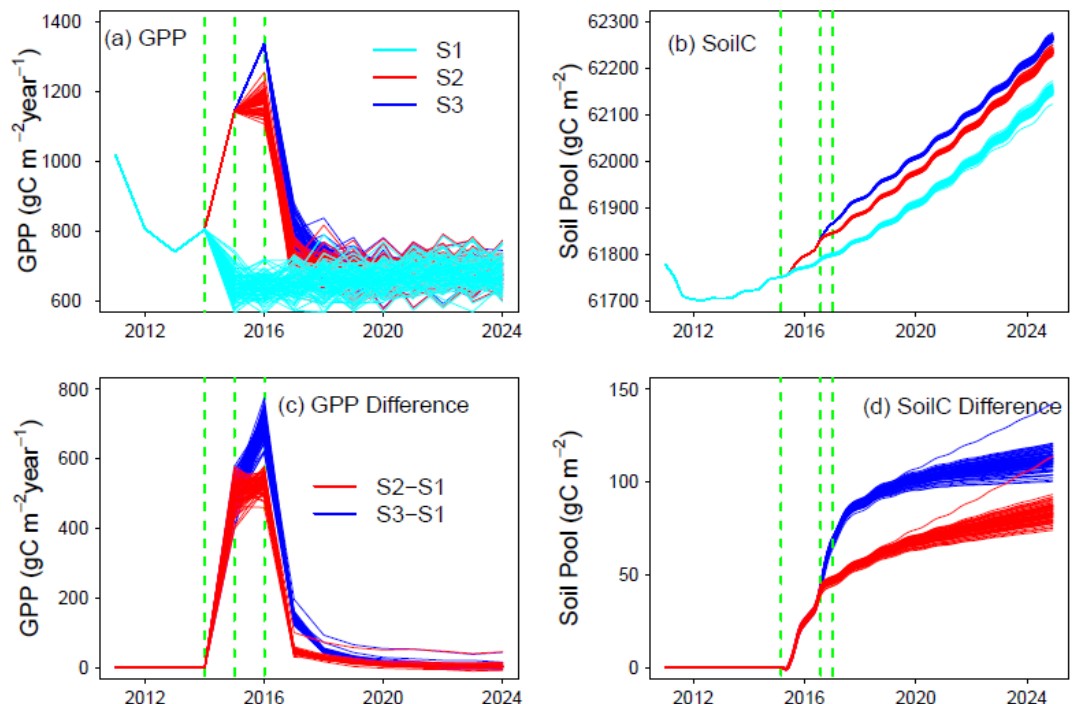
















**Appendices**

**Appendix 1**

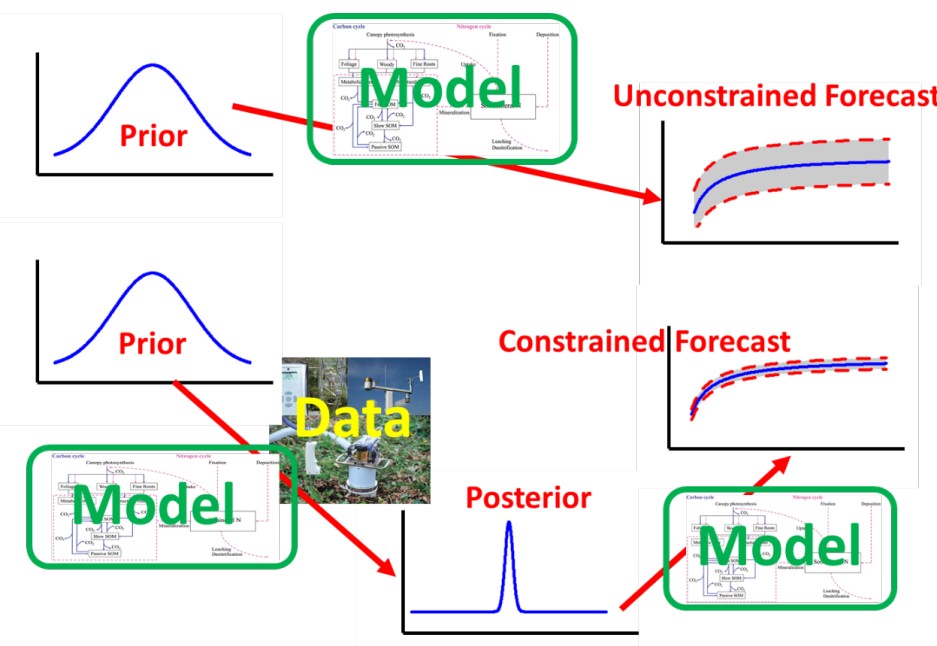

Figure A1. Conceptual demonstration of how data assimilation that updates models through

observations constrains forecasting. The grey shading area corresponds to forecasting

uncertainties.



**Appendix 2 EcoPAD-SPRUCE web portal**


We assimilate multiple streams of data from the SPRUCE experiment to the TECO model using
the MCMC algorithm, and forecast ecosystem dynamics in both near time and for the next 10
years. Our forecasting system for SPRUCE is available at https://ecolab.nau.edu/ecopad_portal/
(the new portal) or http://ecolab.cybercommons.org/ecopad_portal_up/ (the older portal). From
the web portal, users can check our current near and long term forecasting results, conduct model
simulation, data assimilation and forecasting runs, and analyze/visualize model results
(Username: test00 and password:test01 for the new portal; Username: chris and password:chris
for the old portal if login information is required). The login account we created for the new
portal is limited to Simulation only and registration is required for more functionalities.
The main page of the EcoPAD-SPRUCE portal includes animation demos and a brief
description of the system. The animation demos display the dynamic change of gross primary
productivity (GPP), ecosystem respiration (ER), foliage carbon (foliage C), wood carbon (wood
C), root carbon (root C) and soil carbon (soil C) under 10 manipulative warming and elevated
atmospheric $CO_2$ treatments. Each animation shows observations in data assimilation period
during which parameters are constrained (2011-2014) as well as model results (with uncertainty)
from data assimilation and 10 years forecasting from an ensemble of model runs. Warming
generally increase GPP, ER and different carbon pools. Users can also get a sense on how
uncertainties in forcing variables, such as light, temperature, and precipitation that drive carbon
fluxes in terrestrial ecosystem, and limited observations affect uncertainty of GPP prediction.
Under the Custom Workflow menu, users can choose different modes to run TECO model
from the task dropdown box: Simulation, Data Assimilation (DA) and Forecasting (Figure A2).
In the Simulation mode, users are allowed change the initial parameters through "Set Initial



Parameters" button. TECO-SPRUCE currently allows 33 key parameters to be adjustable by
end-users. These 33 parameters include parameters that control soil water dynamics, plant
growth, photosynthesis, carbon allocation among different plant organs, turnover rates of
different pools, temperature sensitivity, and plant phenology. Researchers can choose other
parameters according to their models and specific needs. The simulation runs TECO one time
with user supplied initial parameters and the run normally takes several minutes in the
background. Each requested task from the user is assigned a unique task ID. Users can check
information such as task id, timestamp, parameters, result status, result URL from a web-enabled
report once the task is submitted under the "Task History" tab. If the task status shows
"SUCESS" (Figure A3), users can check datasets relevant to model simulation from the result
URL (for example, http://ecolab.oscer.ou.edu/ecopad_tasks/8b4bcd9b-172c-4031-94b7-
4b080e459025, where "8b4bcd9b-172c-4031-94b7-4b080e459025" is the unique task ID for this
example). The URL directs users to the location (result repository) where information related to
model simulation is stored. Result repository stores parameters supplied to the model run in .txt
format. Yearly and daily simulation results for carbon fluxes and pools are also written in .txt file
format. It also contains .png file format plots of simulated carbon fluxes and pools compared to
observations (Figure A4). Users can check the results from the Task History any time with the
right task ID. With several "Simulation" runs, users can easily get a sense on the sensitivity of
the SPRUCE peatland carbon cycle to different parameters and what are the key processes
regulate the northern peatland carbon dynamics.

Data Assimilation mode enables users to conduct data-model fusion research through a web

portal. A unique feature of the data assimilation portal is that users can pick whatever parameters
to be constrained among the pool of 18 parameters which are important in ecosystem carbon





cycling (Figure A5). Users can change the range of a parameter they are interested in and modify
the initial values of parameters supplied to MCMC. Similarly as in Simulation mode, user can
easily check data assimilation results through the result URL. Results from data assimilation
contain parameter ranges and initial values supplied by users, parameter values accepted in
MCMC, histograms of posterior distribution of parameters (Figure A5), and simulations of
carbon fluxes and pools with 500 randomly chosen accepted parameters. Data assimilation
results are also written into the universal .txt format which makes further utilization of the result
convenient. For example, researchers interested in the pattern and uncertainty in GPP simulation
can quickly get a handle on GPP with an ensemble of easily readable model results.

From the Forecasting mode, users are enabled to set up parameters, or choose posterior

parameters from previous data assimilation results, specify forecast starting and ending dates,
and select warming (0-9 degree Celsius) and $CO_2$ (380-900 ppm) treatments (Figure A6). If a
specific data assimilation result was chosen as input for forecasting simulation, TECO-SPRUCE
would read the constrained posterior parameter file, match the name of constrained parameters to
the whole parameter pool, and then randomly choose 100 sets of constrained parameters to run
forecast. Results from forecast store carbon fluxes and pools from simulations based on the 100
randomly chosen parameters and projected 10 years into the future at the daily time scale. Users
can analyze forecasting dynamics and uncertainties based on stored results. EcoPAD-SPRUCE
result repositories also provide figures that combine observation in data assimilation period,
simulation results in data assimilation as well as forecasting periods, and simulation uncertainty
(Figure A7) to speed up the post-processing of model results.







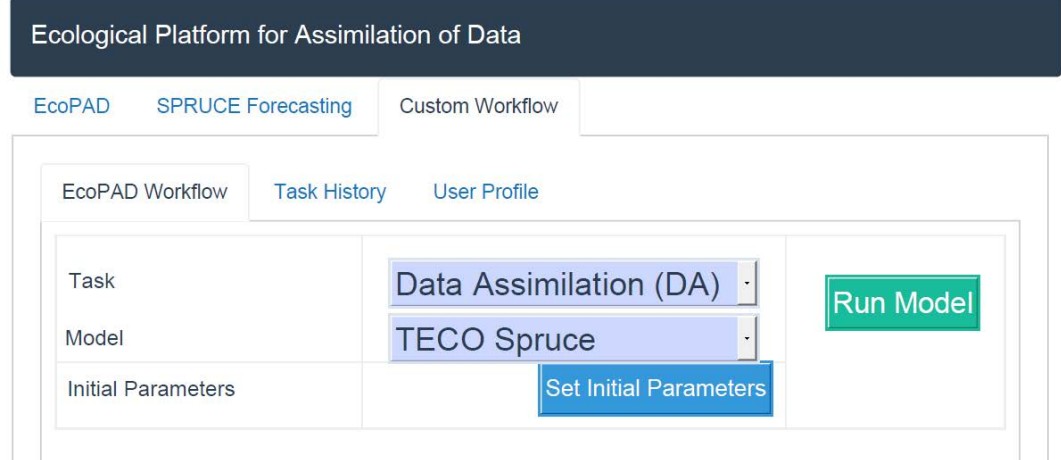



**Figure A2.** The Custom Workflow web portal of the EcoPAD applied for the SPRUCE project.
Users can select among "Simulation", "Data Assimilation (DA)" and "Forecasting" modes from
the task drop-down box to run ecological models in the background. In each mode, users are
allowed to customize the model run, such as set the initial parameter values for "Simulation" and
"Data Assimilation (DA)", choose the updated parameters from "Data Assimilation (DA)" to
conduct "Forecasting" or change the "Forecasting" periods.


Geoscientific Model Development




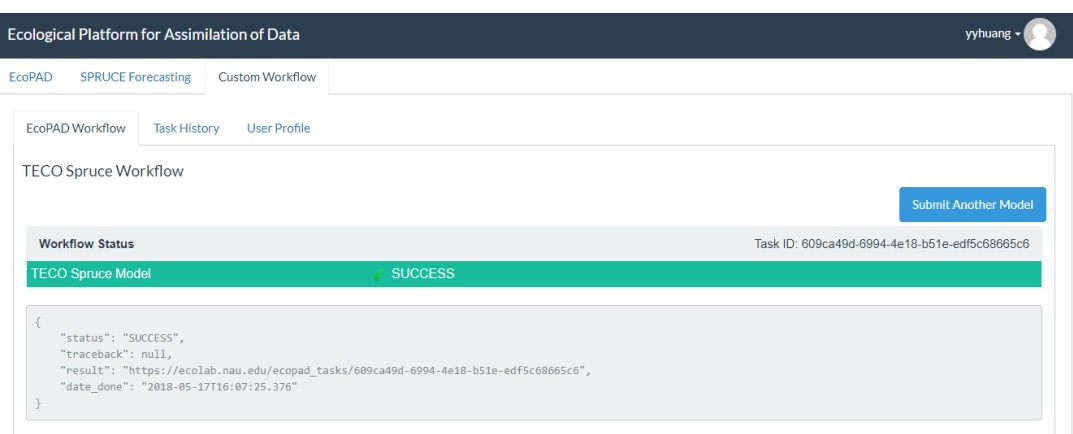



**Figure A3.** An example of a successful model simulations. In EcoPAD, each task is assigned a
unique task ID. The input, output, report and plot relevant to a model task are archived and easy
to tack through the unique web link based on the task ID.

























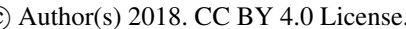




**Figure A4.** An example of the carbon flux and pool size produced from the "Simulation" mode
in EcoPAD-SPRUCE. Red dots indicate available observations and gray lines correspond to
model simulation results. The upper two panels display carbon fluxes: gross primary productivity
(GPP, left panel) and ecosystem respiration (ER, right panel). The lower four panels show result
for foliage carbon (foliage C), wood carbon (wood C), root carbon (root C) and soil carbon (soil
C).

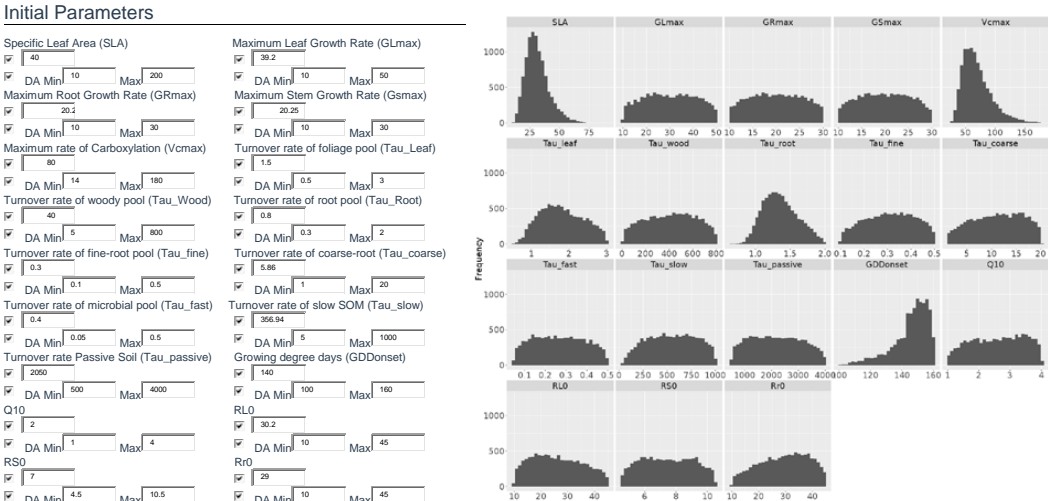


**Figure A5** Parameters that are allowed to modify in EcoPAD-SPRUCE. The left panel shows
the user interface where users can change the initial parameter value and its range supplied to
"Data Assimilation (DA)". The right panel shows the histogram of the posterior distribution of
each parameter that participated in the "Data Assimilation (DA)". The right panel is
automatically generated and archived for each "Data Assimilation (DA)" task.









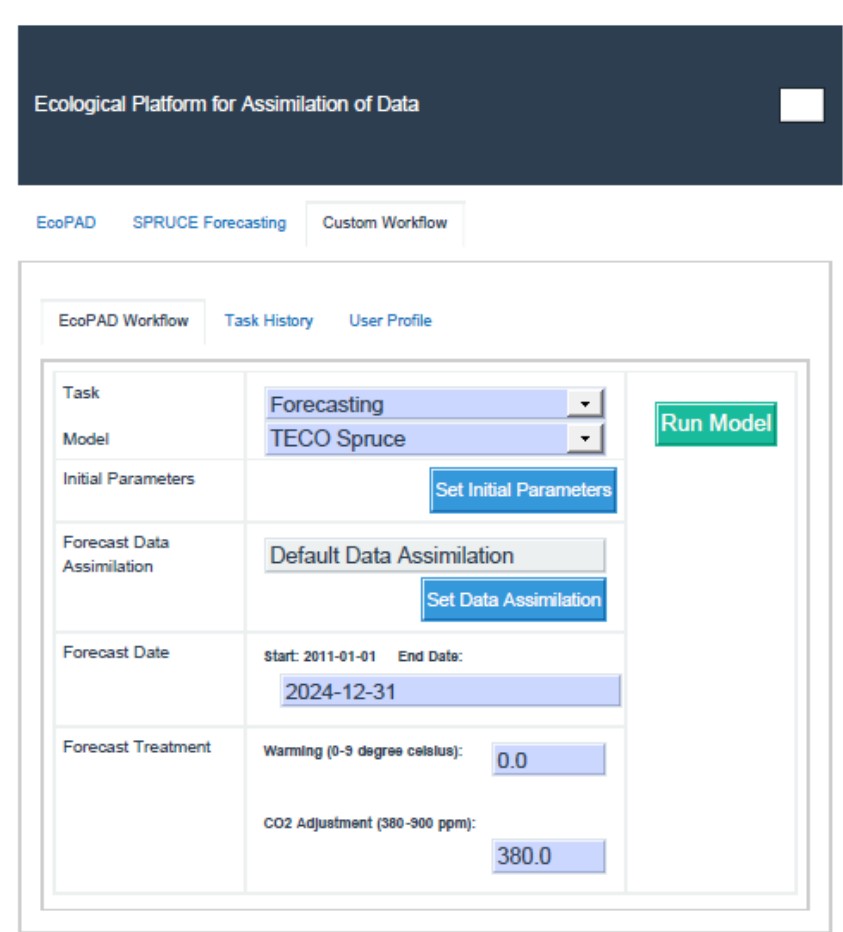



**Figure A6**  An example user interface of the "Forecasting" mode in EcoPAD-SPRUCE.




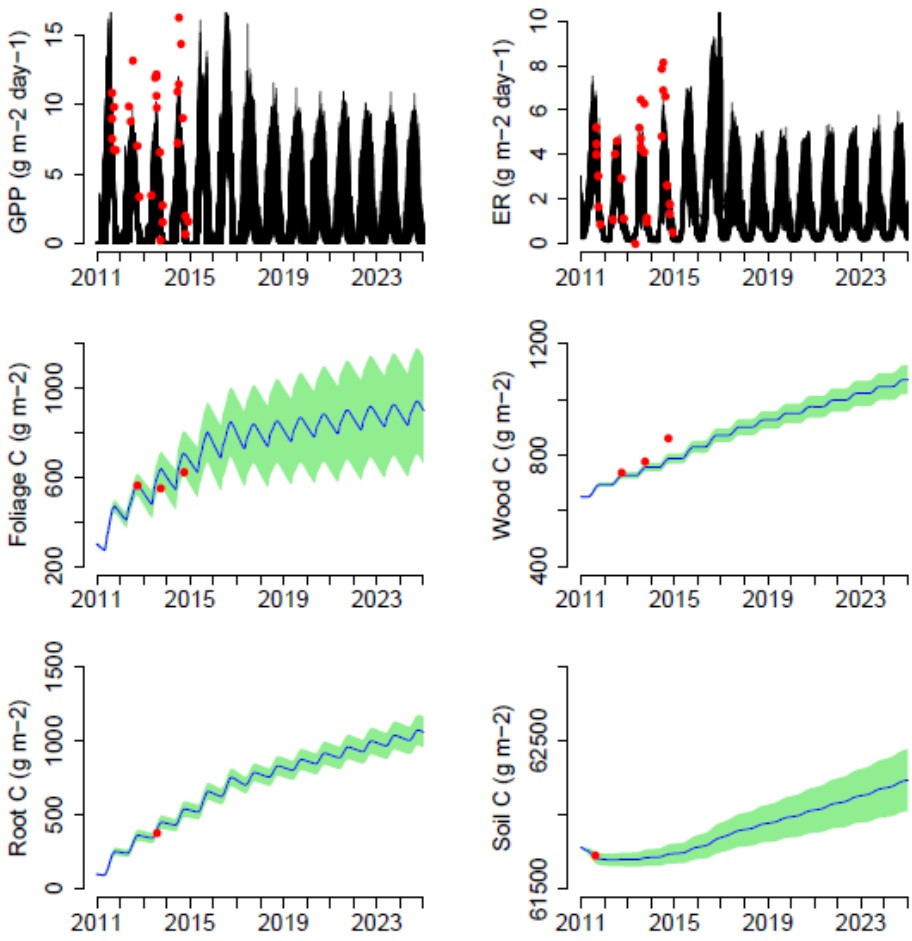


**Figure A7.** An example figure produced from the "Forecasting" mode in EcoPAD-SPRUCE.

Red dots indicate observations used in the data assimilation period (2011-2014). Forecasting

runs from 2015-2024. The upper two panels display dynamic changes of carbon fluxes: gross

primary productivity (GPP, left panel) and ecosystem respiration (ER, right panel). The lower

four panels show result for foliage carbon (foliage C), wood carbon (wood C), root carbon (root

C) and soil carbon (soil C). Blue lines indicate the mean and green shading areas corresponding





to simulation uncertainties for carbon pools generated from an ensemble of model simulations
with randomly chosen parameters from their posterior distributions.


