# Peer review of "Realized ecological forecast through interactive Ecological Platform for Assimilating Data into model (EcoPAD)"

_Geoscientific Model Development, 2018_

## Short Comment (SC1) · 11 Jun 2018

Dear authors, in my role as Executive editor of GMD, I would like to bring to your attention our Editorial version 1.1: http://www.geosci-model-dev.net/8/3487/2015/gmd-8-3487-2015.html This highlights some requirements of papers published in GMD, which is also available on the GMD website in the 'Manuscript Types' section: http://www.geoscientific-model-development.net/submission/manuscript_types.html

In particular, please note that for your paper, the following requirement has not beenmet in the Discussions paper:

[Figure]

- "The main paper must give the model name and version number (or other unique identifier) in the title."

Please provide the version number of EcoPAD in the title of your revised manuscript.

Additionally, please note, that GMD is encouraging authors to provide a persistent access to the exact version of the source code used for the model version presented in the paper. As explained in https://www.geoscientific-model-development.net/about/manuscript_types.html the preferred reference to this release is through the use of a DOI which then can be cited in the paper. For projects in GitHub a DOI for a released code version can easily be created using Zenodo, see https://guides.github.com/activities/citable-code/ for details.

Yours, Astrid Kerkweg

---

## Referee Comment (RC1) · Anonymous Referee #1 · 28 Jun 2018

The manuscript "Realized ecological forecast through interactive Ecological Platform for Assimilating Data into model (EcoPAD)" by Y. Huang et al. presents the development of a web-based software system for quantitative ecological forecasting. The system is based on the availability of observational data, a process-oriented model, an algorithm for assimilating the observations into the model and a web-based workflow. Furthermore the paper describes the application of EcoPAD to the Spruce and Peatland Responses Under Climatic and Environmental change (SPRUCE) experiment in North Minnesota using the Terrestrial ECOsystem (TECO) model and a Markov Chain Monte Carlo assimilation technique in forecasting carbon fluxes and pools.

[Figure]

The manuscript is mostly well written, however, at times (sections 1 and 2) it reads more like a 'sales pitch' for EcoPAD with quite a few repetitive elements (e.g. the list of elements included in the workflow appears at multiple places) and at other times (section 3) it reads as a review on the previous applications of EcoPAD. So in essence, my major concern is that there is little new science in the current version of the manuscript except for the technical engineering of the web-based software system, which in itself is not described in great detail. My recommendation is to focus the manuscript on these technical developments and provide a more in-depth description of the technical details of this system, however, I am not sure if this then still fits to GMD because the web-based software system development is very much focussed on information technology developments.

Another concern relates to the use of the tool by the 'general public' or even experimentalists lacking the background knowledge on data assimilation as promoted by the authors of the manuscript. The concern is that with such a level of automation (essentially only clicking a button on a webpage to get the results of a complex data assimilation experiments) of a very complex system involving experts' concepts from multiple disciplines the user could easily lose the connection to the underlying tools, such as the capability of the ecological model and the data assimilation algorithm. Both components may not be fit for the user's purpose, so a misuse (even and especially unconsciously) of the system can easily happen without the user being able to notice because the user is not an expert of either the ecological model nor the data assimilation algorithm. An erroneous result (which can easily happen if e.g. some observations used in the assimilation are outliers or the assimilation algorithm produces parameter values outside of physical meaningful values etc) of such an automated system could be taken as real and thus be misused. In that sense there should be some caution in promoting this system to non-specialist users.

Detailed comments: Ll 31-33: This sentence is hard to understand, what are updated data?

L40: What is your definition of near real-time?

Ll 67-73: Maybe put a 'e.g.' in front of the mentioned references because these are only examples and there are many more possible references to cite as examples.

Ll 92-94: Unrepresented processes and unknown parameter values are two different reasons for large uncertainties in simulating ecological systems.

Ll 98/99: 'to communicate model with data' seems to be a weird expression.

Ll 122/123 Model improvements do not necessarily happen after the end of an field experiment, other ways of improving a model rely on literature or new theoretical understanding.

L 128: Interactive ecological forecasting does not require web-based technology.

Ll 148/149: This sentence is hard to understand, please clarify what you mean here.

L 175: Do you mean 'quantitative' forecasting?

L 220: Please specify in the manuscript how this is done.

Ll 226/227: It would be interesting to see more details on how the data assimilation system can be independent on the specific ecological model. Usually, in a data assimilation system the underlying model and the applied data assimilation algorithm are closely connected on a code level.

Ll 241-246: Hard to understand, maybe split in two sentences.

Ll 249-252: Again, hard to understand, maybe split in two sentences.

Ll 257-259: The underlying principle of Bayesian modelling is that the ingredients are specified by probability density functions.

Ll 264/265: Complicated formulation, essentially what you want to say is that the posterior uncertainty is smaller than the prior after assimilating observations.

Ll 267-269: Please specify in the manuscript how you choose between DA techniques and what are the criteria for the selection.

Ll 271-273: Again, hard to understand, maybe split in two sentences.

L 275: What are the various uncertainty sources and why do other methods do not take all these sources into account? Please specify in the manuscript.

Ll 296/297: What is a good management in the sense here?

Ll 394/395: What are youngster? And why should they study ecological dynamics through their phones and tables opposed to seniors or others?

L 401: Doesn't that contradict your earlier statement that you need to choose a DA technique that is fit for purpose (Ll 267-269)?

Ll 428-430: How is the automated forecast done? And who is analysing the results of the automated forecast? I suppose if something goes wrong in the automated processing and forecasting an experimentalist won't be able a) notice that something went wrong and b) would be able to fix the bug/problem in the modelling chain.

Ll 443-446: It seems that there is a misconception between parameters and parameterisations: parameters should be invariant in time otherwise they are not parameters but a result of a parameterisation that depends on independent inputs. Could you please clarify this point in the manuscript.

L 500: What are the SPRUCE communities doing with the results?

L 512: 'help experimenters think' is an interesting expression.

Ll 712-714 Could you please clarify this statement. I don't think this is true, complex models can of course assimilate pool-related data, see e.g. Thum et al., 2017.

Ll 729-732: Again, hard to understand, please clarify what you want to say here.

Figure 7: This figure is hard to understand and also the caption doesn't help much

to understand the panels. What exactly has been changed between S1-S3? What is realised and unrealised forecasting? And there seems to be no difference in time-scale among the panels.

Typos: Ll 126, 140, 154, 160, 187, 324, 456, 566

Reference: T. Thum, N. MacBean, P. Peylin, C. Bacour, D. Santaren, B. Longdoz, D. Loustau, P. Ciais: The potential benefit of using forest biomass data in addition to carbon and water flux measurements to constrain ecosystem model parameters: Case studies at two temperate forest sites, Agricultural and Forest Meteorology, https://doi.org/10.1016/j.agrformet.2016.12.004.

―――――――――――――――――――

---

## Referee Comment (RC2) · Anonymous Referee #2 · 22 Aug 2018

This study, "Realized ecological forecast through interactive Ecological Platform for Assimilating Data into model (EcoPAD)," by Huang et al. introduces a web-based platform for data-model integration framework that "can" be used for ecological forecasting. The manuscript introduces the conceptual components of the framework relatively well, albeit with too much generic details on databases, web-based workflow, metadata, data API, which are not the focus of GMDD. Even though I feel that the platform presented in the manuscript has a huge potential, I find the current state of the platform, and the example cases presented are not mature yet (with only one model, one data assimilation, one site). There is a little scientific advance from the study with results based on previous studies. The results and evidences presented, therefore, do not back the

claim of 'have-it-all' platform that can be used by the scientists and 'citizens' alike. I think the manuscript should focus on specifically "what has been done" with thorough scientific discussion, and not "what can potentially be done." This would help judge if the platform is truly flexible and interactive.

The web-based workflow can be viewed as a specific technological advancement in the field of ecological forecasting, but web-based frameworks have been around for a while in the field of geoscience, e.g., PEcAn (as cited in the manuscript in line 138) and PALS. Therefore, I do not agree that it is already the first flexible framework as the manuscript claims. In fact, such claims are not always necessary, but that might just be my personal opinion.

The quality of the figures should be improved, and the redundant information in the schematics should be eliminated. Also, the sources of the images used in the figures are not shown in the respective figures or captions. In general, the schematics can be more technical to suit the expertise of the reader-base of the GMDD.

The manuscript has several Grammatical errors and typos. At times, it feels like even a simple "spelling checker" has not be run through the whole manuscript once. At the same time, some paragraphs are excellently written without a blip.

Major comments:

- The manuscript does not clarify what the "ecological forecasting" means. In the first paragraph, there are several examples of "ecological forecasting" from previous studies. In the end, the EcoPAD seems to be simulating the carbon stocks and fluxes, which is just an aspect of ecological forecasting. The title should be revised to be more specific to the scope and capabilities of EcoPAD.

- There is no specific section on the benchmarking of the performance of the EcoPAD simulations. This is a critical step to have a reliable platform that can be used for forecasting. Therefore, evaluation of model performance should be presented in detail

in this manuscript.

- The examples presented here are for SPRUCE sites. It is not clear whether EcoPAD can be used easily in other sites, even though manuscript ends with statements on how the framework can easily be implemented for different FLUXNET sites and at continental scales. If such implementations are simple as the manuscript claims, they should be the main focus of the manuscript because the results presented here have been, at least partly, used in previous studies.

- Once again, the results presented here just seem like a summary and discussion of previously published manuscripts from the main author and/or co-authors of the manuscript. In fact, I found the results presented in the Appendix A2 to be far more interesting than the results presented in the main text. There should be discussion on why most of the parameters are not well-constrained (Figure A5, right panel), or why Q10 parameters for CH4 is not as well constrained as those for r and why they differ for different temperature treatments (Figure 6). I understand that there may be counter argument on this issue being out of the scope of the current paper, but, it is necessary to discuss how these potentially unconstrained parameters affect the forecast skills of EcoPAD. After all, general public, who do not understand the technical and scientific details, may easily be misinformed with the uncertain forecast of EcoPAD.

- It is not clear from the manuscript what models or assimilation methods are currently available. There are several instances of "ensembles" and "structural uncertainty" but as far as I could decipher, EcoPAD only has one model and one data assimilation method until now. This is also relevant to explaining how EcoPAD can be used to quantify uncertainty from different sources. Such quantification of uncertainty would require factorial experiments with multiple model structures, process/mechanism formulations, cost functions, optimization/assimilation schemes with multiple observational constraints, and so on. It is not clear if EcoPAD already has such functionalities or if it is yet another potential use. If so, an explanation of how "ecologists" can add such functionalities would be useful. For example, is the interested developer responsible

for creating a separate docker that satisfy all the system requirements for his/her own system? I could not test "adding functionality" because it requires registration to the system. Specific comments: Line 53-55: The manuscript does not have any results or discussion on this, and thus this sentence should be removed from the abstract and the whole manuscript.

Line 61: one science - a science?

Line 62: Isn't forecasting always for future?

Line 87-88: what are the "relevant mechanisms" that the previous systems are lacking and how does EcoPAD, and TECO therein, address these shortcomings?

Line 110: one-directionary - unidirectional

Line 114-128: I think the CARDAMOM model-data fusion system (Bloom et al., 2016) deserves a mention in this paragraph (http://www.pnas.org/content/113/5/1285) and in further discussions.

Line 132, 141, 146, 147, 252: spelling errors. I am not mentioning all the places here. Please check the whole manuscript carefully.

Line 151-153: It's not clear what this sentence means.

Line 176: qualitative means better quality or is it quantitative?

Line 210: Should clarify what 'big data' means in this context. Diverse data?

Line 215: cite FLUXNET

Line 305: MongDB - MongoDB

Line 338: May be better to define what IT stands for, just for the sake of completeness.

Line 345: Does docker have a website or citation?

Line 348-350: Isn't it redundant (unnecessary duplication) to include input data in the

docker?

Line 381: I think the web-based platform is for job submission and not web-based simulation.

Line 404: clarify what 'scientific values' means.

Line 415-422: Bayesian statistics has been used in previous ecosystems studies. Please find and cite these previous studies.

Line 433-438: I wonder if experimental/observational setup can be carried out in such short notice.

Line 473: Is +0 the same as control experiment?

Line 479: Is there any difference between data-model integration and data-model communication? If so, this should be clearly stated at the beginning. Both have been used frequently in the manuscript.

Section 3.3: Is this process done systematically or through personal communication? If systematically, how are the updates (both of models and experiments) carried out theoretically and technically?

Line 548-552: In figure 6, it seems like the parameter 'r' is not well constrained for higher treatments of temperature. Discuss the reasons for this.

Line 580: EcoPAD only includes one model, but the sentence says that it relies on ensembles of ecosystem models. This statement is misleading.

Line 586-611: Summary of Jiang et al., 2018 and Huang et al., 2017. So, the results presented in current study are specific for the models and tools used in those previous studies? If there is any additional scientific advancement in EcoPAD, this should be highlighted here.

Line 620-630: To what extent does the parameter ranges depend on the uncertainty

in the forcing? Is there a particular reason why the parameter values are randomly sampled from the posterior? Doesn't it make sense to use the posterior distributions to get the parameter ranges within certain confidence intervals?

Line 631-642: It is not clear if GPP has an effect on carbon stocks in the TECO model.

Line 668: It is not clear how these 'scientific' information is directly useful for general public.

Line 680-681: I am not convinced that all 7 characteristics of EcoPAD have been backed by evidences presented in this manuscript. At least, this has not been clearly presented in the manuscript.

Line 688-705: The discussion here should be divided into the users (those who run the model) and developers (those who add processes and methods to EcoPAD). Since the developers need to carry out a lot of set-up using the GitHub repository, the web-based platform seems more suited to the users. This limits the options of the users to only the ones already available in EcoPAD, which is, as of now, only one model and one data-assimilation system for one site. As such, the potential applications of the model are not applicable to the web-based system. This should be clearly mentioned in the abstract, main text, and the conclusions.

Line 722: 'model structure' - In this use, does it mean different formulations of one process as in Jiang et al., 2018?

Line 744-745: What about the interactions between fluxes and pools?

Line 787-788: Assuming this statement is based on Table 1. But, it is not clear if the table is just a hypothetical example or based on the actual experience.

Line 790-791: I just wonder if it is too risky for experimenters to invest resources on carrying out experiments recommended by modelers who used one-single model?

Line 804-817: I think these tasks of including several sites or using EcoPAD at

continental studies should be a part of this manuscript. As I have mentioned previously, the results presented here have been published in previous studies. Using it in different ecosystems will validate the scientific soundness of EcoPAD and it will provide sufficient evidence of its potential wide-scale applications.

Please also note the supplement to this comment:
https://www.geosci-model-dev-discuss.net/gmd-2018-76/gmd-2018-76-RC2-supplement.pdf

---

## Author Comment (AC1) · 18 Sep 2018

We appreciate the reminder. We add the version number (v1.0) in the revised manuscript.

---

## Author Comment (AC2) · 18 Sep 2018

Anonymous Referee 1 RC1: The manuscript "Realized ecological forecast through interactive Ecological Platform for Assimilating Data into model (EcoPAD)" by Y. Huang et al. presents the development of a web-based software system for quantitative ecological forecasting. The system is based on the availability of observational data, a process-oriented model, an algorithm for assimilating the observations into the model and a web-based workflow. Furthermore the paper describes the application of Eco-PAD to the Spruce and Peatland Responses Under Climatic and Environmental change (SPRUCE) experiment in North Minnesota using the Terrestrial ECOsystem (TECO)

model and a Markov Chain Monte Carlo assimilation technique in forecasting carbon fluxes and pools. The manuscript is mostly well written, however, at times (sections 1 and 2) it reads more like a 'sales pitch' for EcoPAD with quite a few repetitive elements (e.g. the list of elements included in the workflow appears at multiple places) and at other times (section 3) it reads as a review on the previous applications of EcoPAD. So in essence, my major concern is that there is little new science in the current version of the manuscript except for the technical engineering of the web-based software system, which in itself is not described in great detail. My recommendation is to focus the manuscript on these technical developments and provide a more in-depth description of the technical details of this system, however, I am not sure if this then still fits to GMD because the web-based software system development is very much focussed on information technology developments.

Response: We appreciate the Reviewer's valuable suggestions. The manuscript is organized to express the motivation of building EcoPAD, or why we need a platform like EcoPAD in ecological forecasting (Introduction, section 1), the technical support (section 2) and what we can benefit from EcoPAD (or its application and scientific values, sections 2, 3). The technical engineering is an important part of EcoPAD and the manuscript. The functionality of EcoPAD or the role of EcoPAD in advancing ecological forcasting is built upon the technical elements. But the manuscript is not only about technical details. Equally important is what we can benefit from such a platform for ecological forecasting. And the goal of the technical advances is to improve ecological forecasting. We emphasize that iterative interactions between model and data, as well as between modellers and experimenters, are valuable for ecological forecasting.

We do not agree with the Reviewer that there is little new science in the manuscript. The near real time ecological forecasting itself is a new scientific advance in ecology. In addition, we integrated different case studies to illustrate how different components contribute to improve ecological forecasting. Case 3 and Case 4 comes from previous studies. Case 3 is about uncertainty and Case 4 is related to biophysical estimation.

Cases 1, 2, and 5 are new case studies from this manuscript. Case 1 focuses on the communication between modellers and experimenters. We believe that good ecological forecasting is built upon efforts from both modellers and experimenters. Even though this case is not direct technical advance, the techniques embedded in EcoPAD allow near- and real-time interactions between modellers and experimenters. This itself represents an important advance for scientific research that is enabled by modelling. Case 2 is related to acclimation or shift in parameter values. Case 5 compares realised vs. unrealised forecasting. The focus of this study is ecological forecasting. The practice of ecological forecasting is still at its early stage and good forecasting needs to integrate resources from different aspects. Each case study provides valuable information from different perspectives. But none of these cases alone guarantees good ecological forecasting. We keep Case 3 and Case 4 as they reflect important aspects, i.e. uncertainty and boundary conditions, that lead to good ecological forecasting. We have a section discuss the implications of these case studies for better ecological forecasting (section 4.2). And please also refer to our responses to Reviewer 2.

RC1: Another concern relates to the use of the tool by the 'general public' or even experimentalists lacking the background knowledge on data assimilation as promoted by the authors of the manuscript. The concern is that with such a level of automation (essentially only clicking a button on a webpage to get the results of a complex data assimilation experiments) of a very complex system involving experts' concepts from multiple disciplines the user could easily lose the connection to the underlying tools, such as the capability of the ecological model and the data assimilation algorithm. Both components may not be fit for the user's purpose, so a misuse (even and especially unconsciously) of the system can easily happen without the user being able to notice because the user is not an expert of either the ecological model nor the data assimilation algorithm. An erroneous result (which can easily happen if e.g. some observations used in the assimilation are outliers or the assimilation algorithm produces parameter values outside of physical meaningful values etc) of such an automated system could be taken as real and thus be misused. In that sense there should be some caution in promoting this system to non-specialist users.

Response: We agree with the Reviewer that there are risks of misuses. Tool itself does not necessary equal to misuse. It depends on the people who use it and how it is used. Misuse is not unique to web-based simulation and can also occur to non-web-based model simulation and data assimilation. For example, sometimes people who run complex process-based models, such as these embedded in big Earth system models, may not necessary know how different components of the model work. Or an experienced modeler of carbon cycling may not know much about how hydrology in the model works. In these situations, there are also risks of misuse. This is why we emphasize effective communication between different experts. Experimenters may not know the technical details of how to build a model or how to code the data assimilation algorism, but it is not to say they do not need to know how the system works. The communication between modelers and experimenters help the experimenters to understand what works in the background, what is the meaning of a parameter or process, what they can, or cannot do with the platform. The platform is carefully designed to avoid potential errors. For example, the experimenter is asked to prescribe the minimal and maximum values of the parameter they are interested in, avoiding the situation of non-meaningful parameter values. When it comes to outliers in observations or physical/biological boundaries of a parameter, actually, experimenters are more experienced than modellers in making judgements. And normally modellers consult experimenters on the quality and to which degree we can trust and use observation data. The observational data we used in EcoPAD-SPURCE went through the quality control from experimenters. We promote the hands-on experience for the 'general public' with prescribed examples to connect the 'general public' and ecological research. It is not to say we expect the 'general public' to understand the result displayed from the webpage without any guidance or consultancy with a specialist. We still need the modellers to support these activities and play an important role.

Nevertheless, we do not rule out the possibility of potential errors, it is good to be cautious. EcoPAD archives relevant model parameters, boundary conditions, model structure and observational data for each modelling activity. If there are erroneous results, they can be traced through the archives. It does not provide a mechanism to detect unaware erroneous results, but it helps in the situation when people suspect that there are errors.

Detailed comments: Ll 31-33: This sentence is hard to understand, what are updated data?

Response: We changed"updated data" into "new data".

L40: What is your definition of near real-time?

Response: In the SPRUCE study, EcoPAD is setup to automatically update forecasting every week and is adaptable to different updating frequency depending on the research goal. In this specific case, we refer to "weekly" as near real-time.

Ll 67-73: Maybe put a 'e.g.' in front of the mentioned references because these are only examples and there are many more possible references to cite as examples.

Response: Good suggestion. We add 'e.g.'

Ll 92-94: Unrepresented processes and unknown parameter values are two different reasons for large uncertainties in simulating ecological systems.

Response: We agree that unrepresented processes and unknown parameter values can be two different reasons for large uncertainties in ecological modelling. But uncertainty of parameters sometimes also contains information about unrepresented processes. The separation between processes and parameters are context and scale dependent. For example, the decomposition of soil organic matter or litter can be represented through the parameter decomposition rate. The uncertainty of decomposition rate partly reflects unrepresented processes such as microbial dynamics.

Ll 98/99: 'to communicate model with data' seems to be a weird expression.

Response: We change this expression to "to combine model with data".

Ll 122/123 Model improvements do not necessarily happen after the end of an field experiment, other ways of improving a model rely on literature or new theoretical understanding.

Response: We agree that there are other ways to improve model. We add " Data-informed" at the beginning of the sentence.

L 128: Interactive ecological forecasting does not require web-based technology.

Response: We modify the sentence to "The web-based technology facilitates interactions". There are different levels of "interactive", in this manuscript "The interactive feature of EcoPAD (v1.0) is reflected in the iterative model updating and forecasting through dynamically integrating models with new observations, bidirectional feedbacks between experimenters and modellers, and flexible user-model communication through web-based simulation, data assimilation and forecasting." (Lines 191-194, tracked manuscript)

Ll 148/149: This sentence is hard to understand, please clarify what you mean here.

Response: We rewrite this part as "The iterative model-data integration provides an approach to constantly improve ecological forecasting and is an important step especially for realizing near real-time ecological forecasting." And we explained that "Instead of projecting into future through assimilating observations only once, the iterative forecasting constantly updates forecasting along with ongoing new data streams or/and improved models."

L 175: Do you mean 'quantitative' forecasting?

Response: Yes.

L 220: Please specify in the manuscript how this is done.

Response: We add "Each project has a separate folder where data are stored. Data are generally separated into two categories. One is used as boundary conditions for modelling and the other category is related to observations which are used for data assimilation. Scheduled sensor data are appended to existing data files with prescribed frequency." (Lines 254-258, tracked Manuscript )

Ll 226/227: It would be interesting to see more details on how the data assimilation system can be independent on the specific ecological model. Usually, in a data assimilation system the underlying model and the applied data assimilation algorithm are closely connected on a code level.

Response: We agree that there are connections between different components. We added "Linkages among the workflow, data assimilation system and ecological model are based on messaging. For example, the data assimilation system generates parameters that are passed to ecological models. The state variables simulated from ecological models are passed back to the data assimilation system. Models may have different formulations. As long as they take in the same parameters and generate the same state variables, they are functionally identical from the "eye" of the data assimilation system." (Lines 256-262, tracked manuscript )

Ll 241-246: Hard to understand, maybe split in two sentences.

Response: We rewrite this part as "SOM decomposition modelling follows the general form of the Century model [Parton et al., 1988] as in most earth system models. SOM is divided into pools with different turnover times (the inverse of decomposition rates) which are modified by environmental factors such as the soil temperature and moisture."

Ll 249-252: Again, hard to understand, maybe split in two sentences.

Response: We rewrite the sentence as "Data assimilation is growing in importance as the process-based ecological models, despite largely simplifying the real systems, are in great need to be complex enough to address sophisticate ecological issues. These ecological issues are composed of an enormous number of biotic and abiotic factors interacting with each other."

Ll 257-259: The underlying principle of Bayesian modelling is that the ingredients are specified by probability density functions.

Response: It is not clear to us what information the Reviewer intended to add here.

Ll 264/265: Complicated formulation, essentially what you want to say is that the posterior uncertainty is smaller than the prior after assimilating observations.

Response: We agree that what we want to express is that the posterior uncertainty is likely to be smaller than the prior after assimilating observations. We elaborate on this part because some readers of the manuscript might be ecologist/experimentalist with limited background in modelling and Bayesian statistics.

Ll 267-269: Please specify in the manuscript how you choose between DA techniques and what are the criteria for the selection.

Response: Please refer to our response to L 401. EcoPAD is open to different DA techniques.

Ll 271-273: Again, hard to understand, maybe split in two sentences.

Response: We delete "which makes Bayesian inference, especially these with multi-dimensional integrals, workable".

L 275: What are the various uncertainty sources and why do other methods do not take all these sources into account? Please specify in the manuscript.

Response: We remove the statement "is advantageous for better ecological forecasting as it" as it is not the objective of this manuscript to compare different data assimilation techniques.

Ll 296/297: What is a good management in the sense here?

Response: Good management is a subjective term. Nowadays Ecologists are working with large and heterogeneous ecological datasets routinely. Good management can broadly refer to management that improves the efficiency of activities that involve these large and heterogeneous ecological datasets.

Ll 394/395: What are youngster? And why should they study ecological dynamics through their phones and tables opposed to seniors or others?

Response: Youngster is a random example, instead of all-inclusive listing. We use youngster to delegate people who are not experts in ecology. We do not think we intend to have implicit meaning that says seniors or others should not do it. We apologize if we made readers feel in such way. To reduce over interpretation, we replaced youngster with "Non-ecologists, such as youngsters".

L 401: Doesn't that contradict your earlier statement that you need to choose a DA technique that is fit for purpose (Ll 267-269)?

Response: Ll 267-269 states "EcoPAD is open to different data assimilation techniques depending on the ecological questions under study since the scientific workflow of EcoPAD is independent on the specific data assimilation algorithm. For demonstration, the Markov chain Monte Carlo (MCMC) [Xu et al., 2006] is described in this study." We choose a DA technique for demonstration purposes and we do not state that only the chosen DA technique fits. Instead, we think our system is open to different DA techniques and L401 is not in contradiction with our previous statement.

Ll 428-430: How is the automated forecast done? And who is analysing the results of the automated forecast? I suppose if something goes wrong in the automated processing and forecasting an experimentalist won't be able a) notice that something went wrong and b) would be able to fix the bug/problem in the modelling chain.

Response: EcoPAD-SPRUCE is built upon the teamwork. There are both modellers and experimenters. We emphasize the interaction between experimenters and modellers, as illustrated through the section 3.4.1. Modellers built the automated forecasting algorism/code and experimenters also played an important role, such as, in preparing observations and interpretation of the modelling results. Experimenters are not good at finding out software bugs, but they might be more experienced in telling whether the modelling results make sense in reality. Details about how the automated forecast is done can be find in Section 3.3.

Ll 443-446: It seems that there is a misconception between parameters and parameterisations: parameters should be invariant in time otherwise they are not parameters but a result of a parameterisation that depends on independent inputs. Could you please clarify this point in the manuscript.

Response: We think the statement that whether a parameter should be time-independent is context dependent. People commonly link a parameter to a constant that does not change with time. But parameter does not equal to constant. The wiki takes parameter as "A parameter, generally, is any characteristic that can help in defining or classifying a particular system (meaning an event, project, object, situation, etc.). That is, a parameter is an element of a system that is useful, or critical, when identifying the system, or when evaluating its performance, status, condition, etc." (https://en.wikipedia.org/wiki/Parameter). And it is not uncommon to find "time-varying parameters" or "time-variant parameters" in literature, e.g., Tucci 1995, Lauzon and Bates, 1991; Zellner et al., 1991; Zeng et al., 1998; Jiang et al., 2015.

L 500: What are the SPRUCE communities doing with the results?

Response: The results are used mostly for research. From the modelling part, Case 5 (section 3.4.5) is based on this part and ongoing studies are using these archived near-time forecasting to track the time-shift in acclimation and to track model elements that contribute to reducing forecasting uncertainty. The experimenters may adjust their sampling scheme, e.g., the sampling frequency or additional variables to be measured to reduce the forecasting uncertainty.

L 512: 'help experimenters think' is an interesting expression.

Response: We do not understand what the Reviewer intended to express here.

Ll 712-714 Could you please clarify this statement. I don't think this is true, complex models can of course assimilate pool-related data, see e.g. Thum et al., 2017.

Response: The sentence is "In the past, complex models could not assimilate pool-related data to constrain their parameter estimation due to insurmountable computational demand in large scale studies." The context is "large scale studies". Thum et al., 2017 is about site level studies, not large-scale studies. For example, Bloom et al., 2016 assimilated large-scale pool-based observations. So we deleted this paragraph.

Ll 729-732: Again, hard to understand, please clarify what you want to say here.

Response: We replace it with "Parameter values derived under the ambient condition was not applicable to the warming treatment in our methane case due to acclimation".

Figure 7: This figure is hard to understand and also the caption doesn't help much to understand the panels. What exactly has been changed between S1-S3? What is realised and unrealised forecasting? And there seems to be no difference in time-scale among the panels.

Response: The differences between S1-S3 are weather forcings and are indicated by "The upper panels show 3 series of forecasting with updated vs. stochastically generated weather forcing (Lines 1352-1353, tracked manuscript)". We changed "realised" and "unrealised" to "updated" and "un-updated" respectively to reduce confusion. S1 is "un-updated" forecasting and the forecasting is generated with stochastically generated weather forcings over our whole forecasting period (2015-2024). S2 and S3 are updated forecasting. S2 is updated through replacing the stochastically generated weather forcings by measured real weather forcings from January 2015 to July 2016. And S2 then forecasts the period from August 2016 to 2024 with updated states. S3 is updated with measured forcings from January 2015 to December 2016 and forecast

after the end of the real measured forcing. The timing of updating is randomly chosen for demonstration purposes. We added specific time-periods hopefully to make it clear about when measured vs. stochastically generated forcings are used. We also cleared it in the description with " red corresponds to updated forecasting with two stages, that is, updating with measured weather forcing from January 2015 to July 2016 followed by forecasting with 100 stochastically generated weather forcing from August 2016 to December 2024 (S2)" (Lines 1355-1357, tracked manuscript).

Typos: Ll 126, 140, 154, 160, 187, 324, 456, 566

Response: We correct typos throughout the manuscript.

References: Bloom, A. A., J. F. Exbrayat, I. R. van der Velde, L. Feng, and M. Williams (2016), The decadal state of the terrestrial carbon cycle: Global retrievals of terrestrial carbon allocation, pools, and residence times, Proceedings of the National Academy of Sciences of the United States of America, 113(5), 1285-1290, doi:10.1073/pnas.1515160113

Jiang, C., L. H. Xiong, D. B. Wang, P. Liu, S. L. Guo, and C. Y. Xu (2015), Separating the impacts of climate change and human activities on runoff using the Budyko-type equations with time-varying parameters, Journal of Hydrology, 522, 326-338, doi:10.1016/j.jhydrol.2014.12.060

Lauzon, A. M., and J. H. T. Bates (1991), ESTIMATION OF TIME-VARYING RESPIRATORY MECHANICAL PARAMETERS BY RECURSIVE LEAST-SQUARES, Journal of Applied Physiology, 71(3), 1159-1165.

Zellner, A., C. Hong, and C. K. Min (1991), FORECASTING TURNING-POINTS IN INTERNATIONAL OUTPUT GROWTH-RATES USING BAYESIAN EXPONENTIALLY WEIGHTED AUTOREGRESSION, TIME-VARYING PARAMETER, AND POOLING TECHNIQUES, Journal of Econometrics, 49(1-2), 275-304, doi:10.1016/0304-4076(91)90016-7

Zeng, Z., R. M. Nowierski, M. L. Taper, B. Dennis, and W. P. Kemp (1998), Complex population dynamics in the real world: Modeling the influence of time-varying parameters and time lags, Ecology, 79(6), 2193-2209, doi:10.2307/176721 Tucci, Marco P. 1995, Time-varying parameters: a critical introduction. Structural Change and Economic Dynamics, Volume 6, Issue 2, June 1995, Pages 237-260.

Please also note the supplement to this comment:
https://www.geosci-model-dev-discuss.net/gmd-2018-76/gmd-2018-76-AC2-supplement.pdf

**Supplement:**

[revised manuscript text omitted]

Model result

Simulation
Forecast

Experimenter
?

Adjust model
New measurement

Discussion

**Figure 5**

[Figure]

[Figure]

**Figure 6**

[Figure]

[Figure]

---

## Author Comment (AC3) · 18 Sep 2018

This study, "Realized ecological forecast through interactive Ecological Platform for Assimilating Data into model (EcoPAD)," by Huang et al. introduces a web-based platform for data-model integration framework that "can" be used for ecological forecasting. The manuscript introduces the conceptual components of the framework relatively well, albeit with too much generic details on databases, web-based workflow, metadata, data API, which are not the focus of GMDD. Even though I feel that the platform presented in the manuscript has a huge potential, I find the current state of the platform, and the example cases presented are not mature yet (with only one model, one data assimilation, one site). There is a little scientific advance from the study with results based on previous studies. The results and evidences presented, therefore, do not back the claim of 'have-it-all' platform that can be used by the scientists and 'citizens' alike. I think the manuscript should focus on specifically "what has been done" with thorough scientific discussion, and not "what can potentially be done." This would help judge if the platform is truly flexible and interactive.

**Response: We apologize if our manuscript gave the reviewer an impression that EcoPAD is a "have-it-all" platform. As a matter of fact, this manuscript presents the first version of EcoPAD, starting from one model, one data assimilation and its application at one site. Please also refer to our responses to your last two comments below about why we think one model and one detailed long-term site is also important for ecological research. We agree that this version is not the final version of the platform. We add the version number v1.0. In fact, we are currently incorporating the second model into EcoPAD and implementing at two more sites. We hope the functionality of EcoPAD expands as we incorporate more models and more data assimilation techniques into it and it is applied to more sites. Section 4.4 Future developments discusses the future work.**

**Meanwhile, we think this platform is a significant advance in ecological forecasting and should be shared timely with the community to be a benefit from future researches. We appreciate that the Reviewer agrees that this platform has a huge potential in advancing ecological forecasting. Good ecological forecasting relies on integrative and cumulative efforts from multiple sectors of the research community. The work presented here is multi-faceted. It includes the realized near real-time ecological forecasting, the interactive model-experiment system, technical components and specific model elements (model structure, parameter and boundary condition) that affect forecasting. And it involves both modellers and experimenters. The realized near real-time ecological forecasting itself is new and a significant scientific breakthrough in ecological research. The interactive model-experiment system facilitated by EcoPAD is a new paradigm to promote the communication between modellers and experimenters. For the section related to specific model elements, there are 4 case studies, 2 from previous results and 2 from this study. We mentioned 2 cases from previous studies to keep the integrity of the manuscript. But it is not reasonable to assume these 2 cases cover the majority of what we have delivered in this manuscript.**

**We think it is necessary to have the description of the web-based scientific workflow. For one thing, it is relatively new in ecological literature. And on the other hand, the functionality of EcoPAD needs the support of the scientific workflow.**

**We did not claim that the system can be used by the scientists and the citizen alike. We hope our revised manuscript make it clearer. The functionality of EcoPAD is multifaceted. It serves to help ecological forecasting and the priority task of EcoPAD is to improve researches related to ecological forecasting. Meanwhile, the web-based modelling and visualizations help broadly disseminate results of scientific research and extend the service of ecological research to the citizen. Good ecological forecasting need to integrate merits from multiple research communities and be beneficial to the society. EcoPAD is built upon integrating advances from process-based models, observations, data assimilation, information technology and human resources. It incorporates multiple elements, but it is not a "have-it-all" platform.**

The web-based workflow can be viewed as a specific technological advancement in the field of ecological forecasting, but web-based frameworks have been around for a while in the field of geoscience, e.g., PEcAn (as cited in the manuscript in line 138) and PALS. Therefore, I do not agree that it is already the first flexible framework as the manuscript claims. In fact, such claims are not always necessary, but that might just be my personal opinion.

**Response: We agree that the web-based workflow itself has been applied to geoscience for a while. But a platform, such as EcoPAD, that uses the workflow to automate data transfer and processing from sensor networks to ecological forecasting through data management, model simulation, data assimilation, forecasting and visualization is, to the best of our knowledge, among the first. We claimed that the system became the first system to enable interactive model-experiment (ModEx) integration. Based on our knowledge, ModEx is a term that emerged from a workshop organized by Dr. Yiqi Luo in 2012. Although ModEx has been practised for many projects, near-time interactive ModEx was enabled for the first time by EcoPAD. It relies on timely forecasting and bidirectional feedbacks between modellers and experimenters. It works hand-in-hand between modellers and experimenters within the life-cycle of field experimentation, which is not common. Technically, EcoPAD also has its uniqueness. Nevertheless, we agree with the Reviewer that it is not always necessary to claim who is first and we removed such expressions.**

The quality of the figures should be improved, and the redundant information in the schematics should be eliminated. Also, the sources of the images used in the figures are not shown in the respective figures or captions. In general, the schematics can be more technical to suit the expertise of the reader-base of the GMDD.

**Response: We modify Figures 1,3,5 to reduce redundant information and deleted Figure 4. We add description of image sources to the caption of Figure 1: "Images from the SPRUCE field experiments (https://mnspruce.ornl.gov/) are used to represent data collection and the flowchart of TECO model is used to delegate ecological models". Technical details related to Figure 1 are presented through Figures 2 and 3.**

The manuscript has several Grammatical errors and typos. At times, it feels like even a simple "spelling checker" has not be run through the whole manuscript once. At the same time, some paragraphs are excellently written without a blip.

**Response: We go through the manuscript carefully and correct the Grammatical errors and typos.**

Major comments:

- The manuscript does not clarify what the "ecological forecasting" means. In the first paragraph, there are several examples of "ecological forecasting" from previous studies. In the end, the EcoPAD seems to be simulating the carbon stocks and fluxes, which is just an aspect of ecological forecasting. The title should be revised to be more specific to the scope and capabilities of EcoPAD.

**Response: We started the first paragraph with "One ambitious goal of ecology as a science discipline is to forecast states and services of ecological systems. Forecasting in ecology………". Ecological forecasting broadly refers to "Forecasting in ecology". So ecological forecasting covers multiple aspects that ecology covers. We introduced the scientific workflow of EcoPAD that wraps models, observations and data assimilation techniques. We emphasized that the scientific workflow is independent on the specific models. We took the SPRUCE project as an example to illustrate the scientific functionality of EcoPAD. Biogeochemistry is the main focus of the SPRUCE project and the model we wrapped in EcoPAD scientific workflow is a biogeochemical model that simulates carbon stocks and fluxes. No matter which aspect of ecology the model focus on, the functionality and workflow are similar as what we have illustrated through the biogeochemical example. We think the title is reasonable.**

- There is no specific section on the benchmarking of the performance of the EcoPAD simulations. This is a critical step to have a reliable platform that can be used for forecasting. Therefore, evaluation of model performance should be presented in detail in this manuscript.

**Response: Simulations have been evaluated in individual studies. For example, the paper by Huang et al. evaluated biophysical modelling of soil thermal dynamics, snow cover and frozen depths with observations. Jiang et al. (2018) evaluated biogeochemical modelling of carbon pools and fluxes with observations. And Ma et al. (2017) evaluated methane modelling against observation. In the future, we will evaluate accuracy of forecast results and attribute mismatches between forecasts and observations to several sources, such as forcing, model structure, parameterization, and initial values.**

- The examples presented here are for SPRUCE sites. It is not clear whether EcoPAD can be used easily in other sites, even though manuscript ends with statements on how the framework can easily be implemented for different FLUXNET sites and at continental scales. If such implementations are simple as the manuscript claims, they should be the main focus of the manuscript because the results presented here have been, at least partly, used in previous studies.

**Response: We remove the expression of 'easy' throughout the manuscript as it is contextual dependent and the perception differs among people with different backgrounds. Please refer to our initial response related to the main focus and the novelty of this manuscript and to your last comment about expanding this study spatially.**

- Once again, the results presented here just seem like a summary and discussion of previously published manuscripts from the main author and/or co-authors of the manuscript. In fact, I found the results presented in the Appendix A2 to be far more interesting than the results presented in the main text. There should be discussion on why most of the parameters are not well-constrained (Figure A5, right panel), or why Q10 parameters for CH4 is not as well constrained as those for r and why they differ for different temperature treatments (Figure 6). I understand that there may be counter argument on this issue being out of the scope of the current paper, but, it is necessary to discuss how these potentially unconstrained parameters affect the forecast skills of EcoPAD. After all, general public, who do not understand the technical and scientific details, may easily be misinformed with the uncertain forecast of EcoPAD.

**Response: We greatly appreciate reviewer's interest in issues on constraining parameters. It is surely related to the forecast skills of EcoPAD v1.0 as rightfully pointed out by the reviewer. We even more appreciate the reviewer being considerate that the detailed discussion about constraining parameters is "out of the scope of the current paper". It is difficult to balance different elements of the manuscript. For example, Reviewer 1 suggested to focus on the scientific workflow while this Reviewer suggested that the information about the scientific workflow was too much. It is a good idea to dig deep into how not well constrained parameters affect forecasting. The impact of not well constrained parameters is reflected in forecasting uncertainty, which is also an important topic we emphasized in this manuscript. Unconstrained parameters may result in high forecasting uncertainty and therefore low reliability of forecasting result. We added "Not well constrained parameters, for example, caused by lack of information from observational data, contribute to high forecasting uncertainty and low reliability of forecasting results (Lines 797-798, tracked manuscript)." to the section on implications for better forecasting, and also suggested that "….or to what extent unconstrained parameters affect forecasting uncertainty are all valuable questions (Lines 843-844, tracked manuscript)." in the part on forecasting uncertainty.**

- It is not clear from the manuscript what models or assimilation methods are currently available. There are several instances of "ensembles" and "structural uncertainty" but as far as I could decipher, EcoPAD only has one model and one data assimilation method until now. This is also relevant to explaining how EcoPAD can be used to quantify uncertainty from different sources. Such quantification of uncertainty would require factorial experiments with multiple model structures, process/mechanism formulations, cost functions, optimization/assimilation schemes with multiple observational constraints, and so on. It is not clear if EcoPAD already has such functionalities or if it is yet another potential use. If so, an explanation of how "ecologists" can add such functionalities would be useful. For example, is the interested developer responsible for creating a separate docker that satisfy all the system requirements for his/her own system? I could not test "adding functionality" because it requires registration to the system.

**Response: We apologize for the ambiguity. Yes. What presented in this manuscript are based on one model and one data assimilation method. We clarified this point with "Case studies presented in earlier sections are based primarily on one model (Lines 901-902, tracked manuscript)" in the revised manuscript. We also added one paragraph in the future developments section to discuss the concerns raised by the Reviewer.**

**"With these improvements, one goal of EcoPAD is to enable the research community to understand and reduce forecasting uncertainties from different sources and forecast various aspects of future biogeochemical and ecological changes as data become available. The example of Jiang et al. [2018] partitioned forecasting uncertainty from forcings and parameters. An exhaustive understanding of forecasting uncertainty in ecology need to also consider model structures, data assimilation schemes as well as different ecological state variables. Researchers interested in creating their own multiple model and/or multiple assimilation scheme version of EcoPAD can start from the GitHub repository (https://github.com/ou-ecolab ) where the source code of the EcoPAD workflow is archived. To add a new variable that is not forecasted in the EcoPAD-SPRUCE example, it requires modellers and experimenters to work together to understand their process-based models, their observations and how messaging works in the workflow of EcoPAD following the example of EcoPAD-SPRUCE. To add a new model or a new data assimilation scheme for variables that are forecasted in EcoPAD-SPRUCE, researchers need to create additional dockers and mount them to the existing workflow with the knowledge of how information are passed within the workflow."**

Specific comments:

Line 53-55: The manuscript does not have any results or discussion on this, and thus this sentence should be removed from the abstract and the whole manuscript.

**Response: We have examples (e.g., the youngster example and the TreeWatch.Net) and a short discussion (the last paragraph of section 4.4) related to this part. Nevertheless, this part is not the**

**main part of this manuscript and we remove it from the abstract. That being said, we think it is important to make scientific research approachable to the general public.**

Line 61: one science - a science?
**Response: We change "one" to "a"**

Line 62: Isn't forecasting always for future?
**Response: We remove "future"**

Line 87-88: what are the "relevant mechanisms" that the previous systems are lacking and how does EcoPAD, and TECO therein, address these shortcomings?
**Response: The context of "relevant mechanisms" is comparing the non-parametric approach vs. process-based approach in long-term ecological prediction. For example, we can derive the relationship between net primary production (NPP) and light availability based on, say, 10 years' measurement. But to predict NPP of the next 100 years, this empirical NPP-light relationship has limited capacity. The NPP-light relationship may fail to capture the impact of $CO_2$ fertilization or water stress etc. under new conditions. In this case, physiological processes related to NPP coded in process-based models (e.g., the Farquhar photosynthetic scheme) are "relevant mechanisms".**

Line 110: one-directionary – unidirectional
**Response: We change one-directionary to unidirectional**

Line 114-128: I think the CARDAMOM model-data fusion system (Bloom et al., 2016) deserves a mention in this paragraph (http://www.pnas.org/content/113/5/1285) and in further discussions.
**Response: Thanks for suggesting this valuable reference. CARDAMOM is a specific study that applies the data assimilation method. We add it into sections when we mention Bayesian data assimilation and emergent ecological relationships. DART and CCDAS cited here are more about the software environment that makes it easier to conduct data assimilation. And we think it may not be appropriate to cite CARDAMOM here and we also remove the reference to GEMS.**

Line 132, 141, 146, 147, 252: spelling errors. I am not mentioning all the places here. Please check the whole manuscript carefully.
**Response: We correct typos throughout the manuscript.**

Line 151-153: It's not clear what this sentence means.
**Response: We rewrite this part as "Forecasting is likely to be improved unidirectionally in which either only models are updated through observations, or only data collections/field experimentations are**

improved according to theoretical/model information, but not both. Ecological forecasting can also be bidirectionally improved so that both models and field experimentations are optimized hand in hand over time."

Line 176: qualitative means better quality or is it quantitative?
**Response: We change "qualitative" to "quantitative".**

Line 210: Should clarify what 'big data' means in this context. Diverse data?
**Response: We rewrite this part as "The 'big data' ecology generates a large volume of very different datasets across various scales." So the 'big data' refers to both diverse data and the large volume of data.**

Line 215: cite FLUXNET
**Response: We add the reference: Baldocchi, D., E. Falge, L. H. Gu, R. Olson, D. Hollinger, S. Running, P. Anthoni, C. Bernhofer, K. Davis, R. Evans, J. Fuentes, A. Goldstein, G. Katul, B. Law, X. H. Lee, Y. Malhi, T. Meyers, W. Munger, W. Oechel, K. T. P. U, K. Pilegaard, H. P. Schmid, R. Valentini, S. Verma, T. Vesala, K. Wilson, and S. Wofsy (2001), FLUXNET: A new tool to study the temporal and spatial variability of ecosystem-scale carbon dioxide, water vapor, and energy flux densities, Bulletin of the American Meteorological Society, 82(11), 2415-2434, doi:10.1175/1520-0477(2001)082<2415:fantts>2.3.co;2**

Line 305: MongDB – MongoDB
**Response: We correct.**

Line 338: May be better to define what IT stands for, just for the sake of completeness.
**Response: We add "information technology" before "IT".**

Line 345: Does docker have a website or citation?
**Response: We add the docker webpage: https://www.docker.com/**

Line 348-350: Isn't it redundant (unnecessary duplication) to include input data in the docker?
**Response: It is necessary to have the input data in the docker. Each docker is an independent and complete unit that is capable of fulfilling a certain task requested by a user, for example, run a model simulation. This design makes the system easily portable and is not limited by the operation or filesystems, programming language or specific model requirement.**

Line 381: I think the web-based platform is for job submission and not web-based simulation.

**Response: The web-based platform is supported by the scientific flow, observational data, ecological models and data assimilation techniques. It receives requests from the user/command, triggers the task (model simulation, data assimilation or forecasting), carries out the task and displays the results. It is not just for job submissions.**

Line 404: clarify what 'scientific values' means.

**Response: By "scientific values", we refer to the biological, physical or chemical meaning associated with each parameter. We modify "scientific values" to "different biological, physical or chemical meanings".**

Line 415-422: Bayesian statistics has been used in previous ecosystems studies. Please find and cite these previous studies.

**Response: We add the references:**

**Bloom, A. A., J. F. Exbrayat, I. R. van der Velde, L. Feng, and M. Williams (2016), The decadal state of the terrestrial carbon cycle: Global retrievals of terrestrial carbon allocation, pools, and residence times, Proceedings of the National Academy of Sciences of the United States of America, 113(5), 1285-1290, doi:10.1073/pnas.1515160113**

**Ellison, A. M. (2004), Bayesian inference in ecology, Ecology Letters, 7(6), 509-520, doi:10.1111/j.1461-0248.2004.00603.x**

**Jiang, J., Y. Huang, S. Ma, M. Stacy, Z. Shi, D. M. Ricciuto, P. J. Hanson, and Y. Luo (2018), Forecasting responses of a northern peatland carbon cycle to elevated CO2 and a gradient of experimental warming, Journal of Geophysical Research: Biogeosciences, doi:10.1002/2017jg004040**

Line 433-438: I wonder if experimental/observational setup can be carried out in such short notice.

**Response: It depends. As methane is a routinely measured item of the project. If the person is already familiar with methane measurements, one week is enough for preparing. How it operates in practice depends on management. The example here is to show that experimenters can benefit from model information.**

Line 473: Is +0 the same as control experiment?

**Response: This experiment has CO2 fertilization and warming treatments. There are ambient and + 0 °C plots. The difference between ambient and +0 °C treatment plots is the open-topped and controlled-environment enclosure. Ambient plot has no enclosure. We added this explanation to the section related to SPRUCE project. We discard the expression of "control experiment" as it may refer to both.**

Line 479: Is there any difference between data-model integration and data-model communication? If so, this should be clearly stated at the beginning. Both have been used frequently in the manuscript.

**Response:  We do not differ between data-model integration and data-model communication.**

Section 3.3: Is this process done systematically or through personal communication? If systematically, how are the updates (both of models and experiments) carried out theoretically and technically?

**Response: The near real time forecasting is done automatically. However, before setting up the automatic forecasting system, there are extensive non-automatic detailed communication, for example, about the unit of data from sensor vs. model. Experimenters can check forecasting results from the webpage. He or she may adjust the experimental plan, for example, change the date of measurements or make measurements of a new variable. However, the system cannot automatically incorporate measurements of a new variable without additional work of a modeller. The near real time forecasting is automated. But the loop of prediction-question-discussion-adjustment-prediction and benefits from the simultaneous updates of both models and experiments, as we showed in section 3.4.1, need interactive and non-automated communications among modellers and experimenters.**

Line 548-552: In figure 6, it seems like the parameter 'r' is not well constrained for higher treatments of temperature. Discuss the reasons for this.

**Response: Thank you for your comment. From Figure 6, the parameter 'r' was constrained across all treatment temperatures. We calculated the Variance-Mean-ratio (VMR, a value larger than 1 indicates the distribution is constrained) to determine the dispersion of a probability distribution. VMR values for +4.5 °C to +9 °C are 2.1, 2.1, 2.1, 1.7, 1.2, which are all significantly larger than 1.0 based on the t-test. The Reviewer might refer to why the spread or variation of the posterior distribution of the +9 °C treatment is larger than treatments with lower temperatures. The posterior distribution combines information from both the model and data. Neither the model nor the observations are perfect. We have fewer observation data points in higher temperature treatments. And variations from observations are larger in higher temperature treatments. In addition, the model may not be adequate to capture ecosystem responses to extreme temperature changes (i.e., higher temperature changes, e.g., +9 °C and +6.75 °C).**

Line 580: EcoPAD only includes one model, but the sentence says that it relies on ensembles of ecosystem models. This statement is misleading.

**Response: We add "will" to this sentence. And we check throughout the manuscript to correct locations where there could be confusions about what has been done and what will be done. We have a section "future developments" to clarify that multiple models are the future development plan.**

Line 586-611: Summary of Jiang et al., 2018 and Huang et al., 2017. So, the results presented in current study are specific for the models and tools used in those previous studies? If there is any additional scientific advancement in EcoPAD, this should be highlighted here.

**Response: This manuscript focuses on ecological forecasting. EcoPAD is the platform or the tool to help the study of ecological forecasting. We emphasize that integrative efforts are important for better ecological forecasting. The integration relies on advancements from observation, process-based models, data assimilation or parameterization techniques, cyberinfrastructures, human power from both modellers and experimenters etc. We listed 5 cases to illustrate different components that are critical for ecological forecasting and can benefit from the EcoPAD platform. For integrity, we explained studies from Jiang et al., 2018 and Huang et al., 2017. Please also refer to our response to Reviewer 1.**

Line 620-630: To what extent does the parameter ranges depend on the uncertainty in the forcing? Is there a particular reason why the parameter values are randomly sampled from the posterior? Doesn't it make sense to use the posterior distributions to get the parameter ranges within certain confidence intervals?

**Response: Parameter uncertainties (or parameter ranges) are obtained through assimilation observations from 2011 to 2014. In this period, the forcing is the real observed forcing. We do not have complete quantification of measurement uncertainties for each forcing and we did not account for measurement uncertainties of forcing variables. Parameter uncertainties generated in this study come from observational uncertainties of carbon variables.**

**The posterior integrates information from both the prior and observation. It is the best knowledge we can know about parameters. From the posterior distribution, we can get the parameter ranges within certain confidence intervals. However, whether information of parameter ranges alone can be used to derive forecasting uncertainty (or range) depends on complex interactions among parameters, model structures and boundary conditions etc. In non-linear models or there are non-linear interactions among parameters or when the posterior distribution is non-normal, it is not easy to directly propagate parameter range to forecasting uncertainties.**

Line 631-642: It is not clear if GPP has an effect on carbon stocks in the TECO model.

**Response: There is a link between GPP and soil carbon stocks in the TECO model. GPP affects litterfall and therefore the input into soil carbon stocks. As Figure 7 shows, when the difference between GPP is different scenarios (S1, S2, S3) is close to zero, the differences in soil carbon stocks keep growing despite under the same randomly generated forcing. That means, the alternation of soil carbon stocks, no matter it is caused by changed GPP or environmental conditions, affects soil carbon prediction in a longer time scale compared to GPP.**

Line 668: It is not clear how these 'scientific' information is directly useful for general public.

**Response: We remove the "general public".**

Line 680-681: I am not convinced that all 7 characteristics of EcoPAD have been backed by evidences presented in this manuscript. At least, this has not been clearly presented in the manuscript.

**Response: We did not elaborate on these 7 characteristics that are embedded in the system design, especially the workflow. From the previous comment, the Reviewer think it is not necessary to elaborate on the workflow. These characteristics are spread over the scientific workflow section. We do not plan to further elaborate on each characteristic and we do not have to repeat it here. So we removed this sentence.**

Line 688-705: The discussion here should be divided into the users (those who run the model) and developers (those who add processes and methods to EcoPAD). Since the developers need to carry out a lot of set-up using the GitHub repository, the web-based platform seems more suited to the users. This limits the options of the users to only the ones already available in EcoPAD, which is, as of now, only one model and one data-assimilation system for one site. As such, the potential applications of the model are not applicable to the web-based system. This should be clearly mentioned in the abstract, main text, and the conclusions.

**Response: EcoPAD is designed to satisfy the demand of people with different backgrounds. Users of EcoPAD range from people who want to expand and add more components to EcoPAD (developer from the Reviewer's viewpoint) to people who can only use the existing EcoPAD-SPRUCE example. The set-up of GitHub repository is not as easy as using the existing EcoPAD-SPRUCE example, but this is not to say developers do not benefit from such platform. Section 2.3 summarizes how users (including developers) can benefit from the EcoPAD framework.**

Line 722: 'model structure' - In this use, does it mean different formulations of one process as in Jiang et al., 2018?

**Response: Difference in model structure refers to any difference other than parameter values in formulations. It might be formulations of one process or multiple processes.**

Line 744-745: What about the interactions between fluxes and pools?

**Response: It is not clear what this question refers to.**

Line 787-788: Assuming this statement is based on Table 1. But, it is not clear if the table is just a hypothetical example or based on the actual experience.

**Response: The SPRUCE project involves more than 100 scientists with different backgrounds. The discussion started from a teleconference after the delivery of model results, unfortunately, we did not**

**record the teleconference. However, the discussion continued through emails. If necessary, we can show the email communications.**

Line 790-791: I just wonder if it is too risky for experimenters to invest resources on carrying out experiments recommended by modellers who used one-single model?

**Response: Ideally, results or recommendations would be more reliable with multiple models. As a first step, one-single model provides valuable information. We emphasize on the uncertainty of forecasting. Potential results from alternative model structures are likely to be covered, to some extent, by forecasting uncertainty resulted from parameter uncertainties. We also emphasize on the iterative model updates to rely on information from observations. We agree with the Reviewer that one-single model is not the best choice, and it is valuable to incorporate more models in future studies.**

Line 804-817: I think these tasks of including several sites or using EcoPAD at continental studies should be a part of this manuscript. As I have mentioned previously, the results presented here have been published in previous studies. Using it in different ecosystems will validate the scientific soundness of EcoPAD and it will provide sufficient evidence of its potential wide-scale applications.

**Response: We agree with the Reviewer that it is meaningful to expand the application of EcoPAD spatially. We argue that it is equally important to focus on one detailed long-term manipulative ecological study to comprehensively introduce EcoPAD. We chose the SPRUCE experiment as a case to apply EcoPAD partly because the valuable scientific information it provides, and also because the rare opportunities to comprehensively illustrate the functionality of EcoPAD. For example, one of the opportunities is the intensive interactions between modellers and experimenters facilitated by EcoPAD. Both modelling and field experimentation are involved through the life-cycle of the project, which creates the opportunity to illustrate the bidirectional feedback between model forecasting and field experimentation. We are applying EcoPAD to different sites (e.g., precipitation manipulation sites, ecotrons) with different versions of models. However, as a start, we think it is worthwhile to elaborate the technical support and functionality of EcoPAD through EcoPAD-SPURCE.**

**References:**

**Huang YY, Jiang J, Ma S, Ricciuto D, Hanson PJ, Luo YQ (2017) Soil thermal dynamics, snow cover, and frozen depth under five temperature treatments in an ombrotrophic bog: Constrained forecast with data assimilation. Journal of Geophysical Research-Biogeosciences, 122, 2046-2063.**

**Jiang J, Huang Y, Ma S *et al.* (2018) Forecasting responses of a northern peatland carbon cycle to elevated CO2 and a gradient of experimental warming. Journal of Geophysical Research: Biogeosciences.**

**Ma S, Jiang J, Huang YY *et al.* (2017) Data-Constrained Projections of Methane Fluxes in a Northern Minnesota Peatland in Response to Elevated CO2 and Warming. Journal of Geophysical Research-Biogeosciences*, 122, 2841-2861.**

---

## Editor Decision (ED1)

**Review of Huang et al. paper:**

First I sincerely apologize for not having been able to process in due time your manuscript and your detailed responses to the reviewers comments.

After a careful reading of your responses, i believe that you have answered for a large part to the main reviewer's comments. Although one reviewer do not recommend the publication of your paper in GMD, I suggest that it is considered for publication after few last revisions. Indeed, you can still improve the manuscript by taking into account more significantly some of the reviewers critics:

- *Shortening the manuscript*: As pointed by one reviewer, the manuscript has *« quite a few repetitive elements (e.g. the list of elements included in the workflow appears at multiple places) »*. I find that these redundancies are still present and that shortening the manuscript would greatly help. For example information on the potential of EcoPad are presented in the results and then re-state in the discussion section. Also some of the concept brought in the introduction are mentioned also again with similar phrasing in the results or discussion (Ex. 1st paragraph of the discussion section ; 1st paragraph of section 3.4.1 (case 1) is also redundant with the introduction, etc...).
  Please consider decreasing all repetitions that occur in the manuscript in order to make it more concise and thus easier to read. Try to focus the manuscript on what is new by maybe shortening the summary of past experiences described elsewhere. You may also consider grouping the discussion with the results as for some parts the "discussion section" resumes what has been presented in the "result sections".
- *Technical developments of EcoPAD*: As stated by reviewer 1, I also find that an important message of the paper is linked to the "technical implementation of EcoPAD": how generic EcoPAD is, in order to facilitate the inclusion of other models, other experiments and other data assimilation systems. It is not straightforward to relate model state variables to observations for a meaningful data assimilation. I thus agree that more details on the technical engineering could be provided (how it will facilitate the inclusion of other model, data stream, DA,...), possibly in an appendix in order not to overload the core of the manuscript and even if this is slightly in contradiction to reviewer2 suggestions (i.e. that these technical aspects are not the core of GMD). In your response to reviewer1's comment you insist on the scientific messages of the paper; I do agree but these can be more concise and focused on the most novel parts linked to ecological forecasting. To my mind some statements are relatively general and well recognized by the scientific community while the description of how EcoPAD may become a widely used platform is less clear.
- Promotion to non-specialist of EcoPAD: Although your response to reviewer 1 comment is solid, I believe that the discussion section do not emphasize on the limits/risks of web-based tools. No need for large changes but few general warnings/self criticisms could be beneficial.

- Note that reviewer1 concern about your expression ""help experimentaters think" is an interesting expression", is that such expression is quite negative and may imply that experimentaters do not think enough on average!
- Figure 7 (now 6) about updated vs forecasted meteorological impact: Although the new caption and text is more clear, I still find that few more details are needed for a non specialist to understand clearly what is done: what is the updated meteorology and how the stochastically generated forcing is done. Please consider providing few additional information so that the set up of the simulations become clearer.
- Grammar and Typo correction (as pointed by reviewer 2) : although you have clear most of them, I still find some typos or grammatical issues that could be cleared with a thorough reading (ex. P22: "SPRUCE is an ongoing project focuses….")
- Else I do agree that it is difficult to account for some of reviewer2 comments on the need to discuss more in details why some parameters are not well constrained and in the same time to focus the paper on the concept of EcoPAD. Maybe few more self-critical views on the limits of EcoPAD would help.

Best regards,
Philippe